# Digi-Q: Learning VLM Q-Value Functions for Training Device-Control Agents

**Hao Bai** [*]
UIUC

**Yifei Zhou** [*]
UC Berkeley

**Erran Li**
Amazon

**Sergey Levine**
UC Berkeley

**Aviral Kumar**
CMU

## Abstract

While a number of existing approaches for building foundation model agents rely on prompting or fine-tuning with human demonstrations, it is not sufficient in dynamic environments (e.g., mobile device control). On-policy reinforcement learning (RL) should address these limitations, but collecting actual rollouts in an environment is often undesirable in truly open-ended agentic problems such as mobile device control or interacting with humans, where each unit of interaction is associated with a cost. In such scenarios, a method for policy learning that can utilize off-policy experience by learning a trained action-value function is much more effective. In this paper, we develop an approach, called Digi-Q, to train VLM-based action-value Q-functions which are then used to extract the agent policy. We study our approach in the mobile device control setting. Digi-Q trains the Q-function using offline temporal-difference (TD) learning, on top of frozen, intermediate-layer features of a VLM. Compared to fine-tuning the whole VLM, this approach saves us compute and enhances scalability. To make the VLM features amenable for representing the Q-function, we need to employ an initial phase of fine-tuning to amplify coverage over actionable information needed for value function. Once trained, we use this Q-function via a Best-of-N policy extraction operator that imitates the best action out of multiple candidate actions from the current policy as ranked by the value function, enabling policy improvement without environment interaction. Digi-Q outperforms several prior methods on user-scale device control tasks in Android-in-the-Wild, attaining 21.2% improvement over prior best-performing method. In some cases, our Digi-Q approach already matches state-of-the-art RL methods that require interaction. The project is open-sourced at https://github.com/DigiRL-agent/digiq

## 1 Introduction

Foundation models (OpenAI, 2024a; GeminiTeam, 2024) open up the possibilities to build agents that make intelligent decisions in the real world (Liu et al., 2023). While prompting off-the-shelf language models with specific instructions is one way to get them to make decisions, this is not good enough for attaining goals and maximizing rewards that are critical in downstream tasks (Zeng et al., 2023; Chen et al., 2023). Part of the reason is the lack of sufficiently diverse decision-making data (Gur et al., 2023). But perhaps a more fundamental challenge is that simply imitating Internet data is not good enough for training models "how" to act intelligently, reduce uncertainty, and achieve goals in non-stationary real-world decision making settings (Bai et al., 2024; Ma et al., 2024).

Recently, the community has been turning towards using reinforcement learning (RL) for training agentic policies. RL avoids the shortcomings of imitation and prompting, by explicitly training the policy to solve tasks (Zhou et al., 2024b; Verma et al., 2022; Snell et al., 2023; Abdulhai et al., 2023). That said, the best performing RL methods today for improving a policy in multi-step agentic tasks rely critically on interaction due to the use of policy gradient updates (Yao et al., 2023) coupled with Monte-Carlo values (Bai et al., 2024; Putta et al., 2024; Shao et al., 2024), which often require sufficient amounts of on-policy data to get a low-variance learning signal. The amount of on-policy data needed is likely only larger in non-stationary and dynamic environments (Bai et al., 2024).

---

[*]Equal Contribution, correspondance to: haob2@illinois.edu, yifei_zhou@berkeley.edu

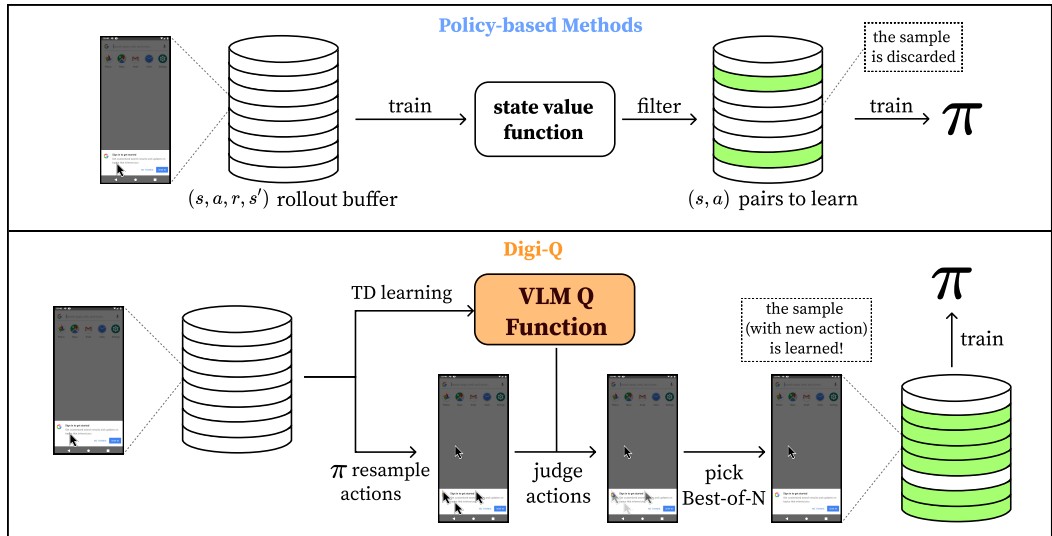

Figure 1: **Comparing Digi-Q with on-policy policy-gradient methods.** $(s, a)$ rollout pairs that are learned are marked green in the buffer. Typically, policy-based methods utilizes a state value function to filter out promising state-action pairs, and requires online data to improve. In contrast, Digi-Q learns a state-action ($\mathbf{Q}$) value function through TD-learning on offline data, and re-sample an amount of actions for each state. This Q-function is then used to rank the re-sampled action to learn a policy using the best action under each state. Digi-Q results in higher sample efficiency than policy-based methods to be applied even in a fully offline setting.

If on the other hand, we could train a critic (i.e., an *action-value function*) that could score a policy's action reliably, *without* needing to actually simulate the policy behavior over multiple steps several times, we could simplify our recipe for policy improvement without costly simulations. With this motivation, in this paper, we build a simple approach to train a VLM Q-function. In our problem setting of mobile device control, this corresponds to training a VLM-based Q-function that can provide a score for every snapshot of the phone screen and an action, represented by laying the cursor over this snapshot. Our method, Digi-Q, trains a Q-function for device control entirely using historical data collected by (potentially suboptimal) agents.

To train Q-functions effectively, Digi-Q handles a number of challenges posed by value learning at scale: **(i)** instability in running temporal-difference (TD) learning for training value functions with large models (Kumar et al., 2021) and **(ii)** inefficiency of TD backups per unit amount of "compute" (i.e., gradient steps spent) (Chebotar et al., 2023) (see Appendix B.1). Digi-Q does so by training on top of a frozen intermediate layer *representation* of the VLM instead of training all parameters of the VLM. For attaining the best performance, representations from off-the-shelf VLMs are not good enough, since they often do not contain feature information crucial for predicting actions or their consequences. Therefore, Digi-Q prescribes running an initial phase of representation fine-tuning to prime representations of a VLM to be more amenable to TD-learning.

To specifically understand benefits of learning the Q-function, we note that while state-only Monte-Carlo value functions (Bai et al., 2024; Zhai et al., 2024) can only evaluate the efficacy of a single action that was actually executed at any given state, a good Q-function provides us with reliable estimates of the future expected reward for *multiple* actions at the same state, without needing to actually roll these candidates out. This allows us to develop a Best-of-N policy-extraction objective that trains the agentic policy to imitate the best-rated action per the Q-function without any additional interaction. The difference between policy-based methods and Digi-Q is illustrated in Figure 1.

The main contribution of this work is Digi-Q, an offline approach to train a VLM-based Q-function for building device-control agents. Digi-Q represents and trains Q-functions on top of intermediate representations from a VLM, fine-tuned to especially consist of actionable information. Digi-Q unlocks the use of a Best-of-N policy extraction objective to make the most effective use of a Q-function in obtaining a policy. The agent produced by running Digi-Q on offline data outperforms prior approaches that also extract policies from offline data in the problem setting of Android device control (Rawles et al., 2023) with 21.2% of relative improvement over the best-performing prior

method, even though these domains remain challenging for state-of-the-art proprietary models (Liu et al., 2024c; Bai et al., 2024). *To the best of our knowledge, this work is the first to successfully scale state-action Q-value functions to realistic agent tasks with VLMs.*

## 2 RELATED WORK

**RL for training GUI and device-control agents.** Due to their reasoning and perception capabilities, LLMs and VLMs have been applied extensively to build agents to navigate web pages (Zhou et al., 2024a; Koh et al., 2024a; Deng et al., 2023; Zheng et al., 2024; He et al., 2024) and GUI interfaces (Bai et al., 2024; Yan et al., 2023; Hong et al., 2023; Rawles et al., 2023; 2024; Zhang and Zhang, 2024). In contrast to using off-the-shelf proprietary models (Zheng et al., 2024; Yan et al., 2023; Zhang et al., 2023; He et al., 2024) or fine-tuning them with a small amount of human demonstrations (Hong et al., 2023; Zhang and Zhang, 2024; Zeng et al., 2023), RL provides the advantage of optimizing task-specific reward and goal-oriented behavior, which is important in dynamic and non-stationary environments especially when human demonstrations are stale (Bai et al., 2024; Zhou et al., 2024b; Putta et al., 2024; Pan et al., 2024; Song et al., 2024). However, most successful applications of RL for real-world GUI agent tasks use less efficient algorithms such as (nearly) on-policy policy gradient or filtered imitation learning (Bai et al., 2024; Putta et al., 2024; Song et al., 2024; Koh et al., 2024b; Shao et al., 2024). This can be problematic for real-world GUI agent tasks where interaction with the actual environment presents a bottleneck and on-policy data is hard to collect due to practical issues such as privacy. In traditional RL, the approach to avoid variance and learn without on-policy interaction (or massive simulation) is to actually train a Q-value function that can score a given action at a given state (snapshot of the phone screen). Using this Q-value function for training the policy results in substantially better performance (Mnih et al., 2013; Haarnoja et al., 2018; Fujimoto et al., 2018) and can be done entirely from historical data (Kumar et al., 2020; Fu et al., 2021). To the best of our knowledge, our work is the first to scale value-based Bellman backups to convert VLMs into device-control Q-functions, which serve as effective scoring functions to policy extraction.

From an algorithmic standpoint, the closest work to ours that trains agents with Q-functions is ArCHer (Zhou et al., 2024b), which builds a hierarchical framework for developing RL algorithms for training agents. Note that this prior work presents results on simplified environments (Yao et al., 2023). While our use of a VLM-based value function to train the policy can be interpreted as yet another algorithm under the hierarchical actor-critic abstraction in ArCHer, note that the methodology for running RL at scale is substantially different from this prior work. Specifically, we prescribe several important components along the use of frozen VLM representations and a policy extraction approach based on best-of-N policy extraction, which also enables scaling test-time compute. Our experiments in Section 5 show that Digi-Q is much more effective (about a 20% improvement as shown in Section 5.2) than the policy-gradient algorithm used by Zhou et al. (2024b). This justifies the benefits of our seemingly simple, yet an effective design of using the value function. Other works (Zhai et al., 2024) train VLMs with on-policy PPO (Schulman et al., 2017; Chen et al., 2024). Finally, Chen et al. (2024) runs RL on top of frozen VLM representations as well, although unlike us they do not fine-tune the VLM to make these representations more amenable for fitting value functions. Instead, they use handcrafted prompts to prime the VLM into producing useful features. Our results highlight that the representation fine-tuning phase in Digi-Q is critical to obtaining a good Q-function, but prompting alone is not as effective.

**Challenges of training an off-policy Q function with foundation models.** Despite the efficiency and data reuse benefits of training a Q function via off-policy TD-learning, it can be unstable and computationally inefficient if not treated carefully, particularly the case for large foundation models with billions of parameters. This instability stems from two aspects: (**1**) prior work has often found it hard and unstable to train value functions via Bellman backups and TD-learning (Kumar et al., 2021; 2022; Chebotar et al., 2023), which is challenging at scale. To address this, Chebotar et al. (2023) had to employ a combination of conservative regularization (Kumar et al., 2020) and regularization with n-step returns (Hessel et al., 2018) resulting in a complex approach; (**2**) policy extraction from trained Q-functions often utilizes policy gradient approaches with a "negative gradient" term (Tajwar et al., 2024) that can be unstable with offline data. This has largely resulted in the community focusing on on-policy or filtered imitation learning methods. However, Park et al. (2024) show that supervised regression methods such as AWR (Peng et al., 2019) can lead to slow convergence and poor asymptotic performance. To address challenge (**1**), Digi-Q runs TD-learning on top of frozen VLM representations, but after a fine-tuning phase to make them more amenable to representing Q-

functions and to address **(2)**, we introduce a Best-of-N based policy extraction loss, akin to concurrent work (Mark et al., 2024) in the domain of robotic learning.

## 3 PRELIMINARIES AND PROBLEM SETUP

We aim to build value functions for training agents in the domain of device control, where we wish to accomplish pixel-based interactions on virtual devices, following similar protocol as past work (Bai et al., 2024). In this section, we will discuss the setup for this problem, followed by terminology, notation, and background that would be useful in developing our approach in the next section.

### 3.1 PROBLEM SETUP: ANDROID DEVICE CONTROL

We scope our study in the domain of pixel-based Android device control (Bai et al., 2024; Zhang et al., 2023; Rawles et al., 2023). Each episode in this domain starts with a fully-functioning Android emulator reset to the home screen, and a task is randomly drawn from a task pool represented by natural language instructions. The agent needs to complete the task through pixel-based interactions with the device as illustrated in Figure 1. The actions that the agent can take are primitive pixel-level commands such as clicking at a coordinates and typing text. Concretely, given a screenshot of a phone, we want the agent to output a string command such as "click (0.8, 0.2)" to be executed in the environment at the current step, where 0.8 and 0.2 are 0-1 normalized x-y coordinates in the screen. This domain is known to be more general and challenging than web navigation alone or link-based device control (Bai et al., 2024), and present many real-world challenges of device stochasticity and dynamism, such as unpredictable distractors like pop-ups and technical glitches like incomplete website loading. Following Pan et al. (2024); Bai et al. (2024), the agents are evaluated via binary 0/1 rewards from a proprietary model (i.e., Gemini 1.5 Pro (GeminiTeam, 2024)) that makes a verdict of whether the specific task has been completed at each step. We want to use this signal to learn a value function that can accurately predict the future outcome that the agent would attain if it were to execute a given action at a given snapshot, without actually executing this rollout. More importantly, we wish to learn this value function using a static dataset $\mathcal{D}$ storing historical past interaction data and use it to learn an agentic policy.

### 3.2 REINFORCEMENT LEARNING DEFINITIONS

There are two types of value functions we can model in our setting: **(1)** a "turn-level" value function that can score each natural language interaction with the external environment (e.g., "type box [2]: wikipedia of chocolate"), and **(2)** a value function at the "token-level", where each step is an independent natural language token. Terminology wise, we define the **state** $s_t$ in the Markov decision process (MDP) in our setting to be the sequence of tokens denoting the log of the interaction history of the agent with the environment thus far concatenated to the current observation. Each **action** $a_t$ is a sequence of tokens directly applied to interact with the environment at the entire turn level.

The turn-level Q-function for a given policy $\pi$ is the expected cumulative reward of a particular action at the current step, and then following the policy $\pi$ thereafter: $Q^\pi(s_h, a_h) = \mathbb{E}_\pi \left[ \sum_{t=0}^\infty \gamma^t r(s_{h+t}, a_{h+t}) \right]$. The value function of a policy $\pi$, $V^\pi(s_h)$, is defined as the expected Q-value, $Q^\pi(s_h, a_h)$, where actions $a_h$ are sampled from the policy $\pi$. The advantage function $A^\pi(s_h, a_h)$ corresponds to the relative benefit of taking action $a_h$ in state $s_h$, and is computed as the difference between the Q-value and the value of the state under the policy: $A^\pi(s_h, a_h) = Q^\pi(s_h, a_h) - V^\pi(s_h)$. The goal of RL is to train a policy that can produce token sequences that maximize discounted cumulative rewards over the course of a rollout.

For training the agentic policy, our approach seeks to train an action-value Q-function $Q$ parameterized by parameters $\theta$, and a policy parameterized by $\phi$. Additionally, we maintain a state-only value-function $V$ parameterized by $\psi$ to stabilize training. Both Q- and V- functions are instantiated by a small MLP layer on top of a VLM backbone. We use $\theta_{\text{VLM}}, \psi_{\text{VLM}}$ to represent the parameters of the VLM backbone and similarly $\theta_{\text{MLP}}, \psi_{\text{MLP}}$ for parameters of the MLP head. We will denote the last layer representations of these VLM backbones as $f_{\theta_{\text{VLM}}}(s, a)$ and $f_{\psi_{\text{VLM}}}(s)$.

### 3.3 BACKGROUND: ARCHER FRAMEWORK FOR TRAINING AGENTIC POLICIES WITH RL

The ArCHer framework (Zhou et al., 2024b) provides one conceptual way to pose training of foundation model value functions and agentic policies as a hierarchical RL problem. Although their framework is not specific to one particular RL algorithm, Zhou et al. (2024b) show that a convenient

way to instantiate this approach is to learn a value function at the turn-level and a policy to produce tokens in an autoregressive manner. The value function critic and the agentic actor are then optimized against each other similarly to standard actor-critic RL. The loss functions for training the cirtic are given by:

$$J_Q(\theta) = \mathbb{E}_{s,a,r,s'\sim\mathcal{D}} \left[ (Q_\theta(s,a) - r - \gamma V_{\bar{\psi}}(s'))^2 \right]. \tag{1}$$

$$J_V(\psi) = \mathbb{E}_{s\sim\mathcal{D}} \left[ \mathbb{E}_{a\sim\pi_\phi(\cdot|s)} \left[ (V_\psi(s) - Q_{\bar{\theta}}(s,a))^2 \right] \right]. \tag{2}$$

$\bar{\theta}$ and $\bar{\phi}$ are the delayed target networks (Mnih et al., 2013) for stability and they are updated as an exponential moving average of $\theta$ and $\phi$. The instantiated algorithm from Zhou et al. (2024b) supports policy extractions through REINFORCE policy gradient:

$$J_\phi(\pi) = \mathbb{E}_{s_c\sim\mathcal{D},a_t^{1:L}\sim\pi_\phi} \left[ \sum_{i=1}^{L} A(s_c,a_t) \log \pi_\phi(a_t^i|s_c,a_t^{1:i-1}) \right].$$

While our approach will utilize a similar framework to conceptualize the training of the value function critic and the agentic policy in our method, the design of actor and critic updates employed are substantially different in our method. As such, the actor update from Zhou et al. (2024b), which also corresponds to the standard policy gradient update, can also be unstable in certain problems.

# 4 DIGI-Q: TRAINING VLM Q-VALUE FUNCTIONS FOR AGENTIC POLICY

To obtain an effective agentic policy for device control problems, Digi-Q trains a Q-value function on static data, which is then used to extract a policy. In the process of designing Digi-Q, we need to address challenges with running value-based RL at scale. First, to avoid pathological behavior of TD-backups with large models (Zhou et al., 2024b; Snell et al., 2023; Abdulhai et al., 2023; Chebotar et al., 2023) and to avoid the computational costs associated with training a billion-parameter VLM end-to-end with TD-learning, we train Q-functions on top of frozen VLM representations. Since VLMs are not trained on substantial quantities of decision-making data, off-the-shelf VLMs largely do not accurately represent *actionable* elements of an input scene. To address this, Digi-Q fine-tunes VLM representations before running Q-function training. This fine-tuning procedure is not the same as supervised fine-tuning on in-domain data, but rather is designed to emphasize actionable features that change from one snapshot to the other and hence help model the value function.

Unlike typical on-policy RL or filtered imitation learning (e.g., AWR (Peng et al., 2019)) that only updates the policy with one action per state, training a Q-function allows us to simultaneously estimate returns from multiple action candidates, all of which can then be used for improving the policy. Using multiple action candidates can be more efficient, especially if the critic predictions are more liable, and even if not, it offers variance reduction benefits. Digi-Q utilizes this insight to develop a Best-of-N reranking based policy extraction objective for training the policy. This policy improvement operator is stable and more effective than policy gradient or advantage-weighted regression in our expriments. The method is illustrated in Figure 2, with each part described below.

## 4.1 TRAINING VLM Q-FUNCTIONS VIA TD-LEARNING AND REPRESENTATION FINE-TUNING

As discussed above, fine-tuning large VLMs end-to-end to represent value functions can present pathologies and perhaps a more practical approach is to train a separate value function on top of the frozen representation from the VLM. However, most VLM backbones are largely not trained to model *actionable* information for a given scene or make predictions about possible scenes that could result from taking actions in an environment. If the internal representations of the VLM do not pay attention to actionable information in a scene correctly, then training a state-action Q-function $Q(s,a)$ will either degenerate into learning a state-only value function $V(s)$ (i.e., it will ignore the action input) or diverge by incorrectly amplifying noise in partially-trained Q-value estimates at unseen, out-of-distribution actions that appear in the right hand side of a TD-backup during training. Indeed, in our preliminary runs, we found that while open-source VLMs such as LLaVa-1.5 (Liu et al., 2024a) are able to answer questions about a scene, the same VLM does often fail at correctly answering questions about impact of current actions into the future, e.g., it answers "does this clicking operation lead to a new page in ebay.com?" incorrectly.

To address this issue with VLM representations, Digi-Q first fine-tunes representations of a VLM with an binary classification objective to enable it to pay attention to actionable features of an input

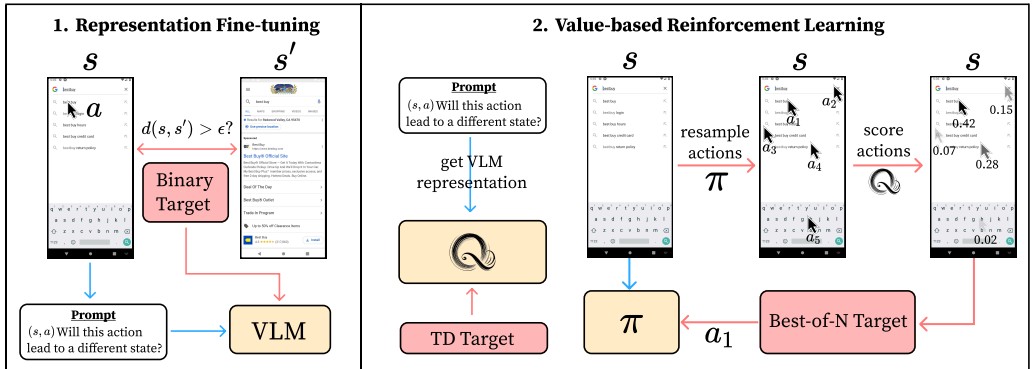

Figure 2: **Overview of Digi-Q.** Blue arrows represent forward data flows, while red arrows represent how we get learning targets used for back propagation. Our method first goes through a representation fine-tuning stage to extract actionable features from the VLM. TD-learning is then performed on top of frozen VLM representations to learn a reliable Q value function, followed by Best-of-N policy extraction approach.

scene. Once fine-tuned, the representations of this VLM are used to train a Q-function represented using a small MLP on top of the frozen representation. Not only is this approach more stable and robust against pathologies that could emerge from fine-tuning, but it cuts down computational costs since only 1% of all the VLM parameters (the head) are now trained via TD-learning.

**Approach.** Our representation fine-tuning objective is constructed as follows: given a transition pair $(s_t, a_t, s_{t+1})$ drawn from a replay buffer, the representation fine-tuning objective attempts to model if and how the next state $s_{t+1}$ will change from the current state and action $(s_t, a_t)$. Our observation is that in device control problems, a useful action should lead to a substantial visual change in the pixel values of a scene (e.g., successfully typing a search query and pressing enter on "google.com" should cause the page to change substantially to now show a list of search results, whereas an unsuccessful search attempt, say due to imperfect clicks, with a very high probability will change none to few pixels of the original scene and remain stuck on the "google.com" page ). Equipped with this insight, we construct positive and negative tuples of transitions $(s_t, a_t, s_{t+1})$, where the positive tuples consists of transitions change the state significantly (i.e., larger than a threshold $\epsilon$ on the $\ell_2$ image distance) and the negative tuples are the remaining transitions. This is equivalent to assigning a binary $\{0, 1\}$ label to a transition:

$$y_t = \begin{cases} 0, & d(s_t, s_{t+1}) < \epsilon \\ 1, & \text{otherwise} \end{cases}$$

Now, the VLM is trained to produce this 0-1 label $y_t$ given a state-action tuple $(s_t, a_t)$ using a binary cross-entropy loss on its parameters $\theta_{\text{VLM}}$:

$$J_{\mathcal{P}}(\theta_{\text{VLM}}) = -\mathbb{E}_{s_t, a_t \sim \mathcal{D}}[y_i \log \mathcal{P}_{\theta_{\text{VLM}}}('\text{yes}'|s_t, a_t) + (1 - y_i) \log \mathcal{P}_{\theta_{\text{VLM}}}('\text{no}'|s_t, a_t)], \quad (3)$$

where $\mathcal{P}_{\theta_{\text{VLM}}}$ is the next-token distribution obtained from the VLM backbone.

After this phase of representation fine-tuning, we freeze the parameters of VLM, and extract the embedding of the yes/no token output to serve as the input representation of $(s_t, a_t)$ to the Q function. We now run a TD-learning objective from Equation 1 in Section 3 to train the Q-function.

Note that Equation 1 also utilizes a parameterized value function. Since the value function does not depend on the action, we are able to directly use internal representations of an off-the-shelf VLM, without requiring any phase of initial fine-tuning. On top of the frozen representations from the VLMs, our value functions $\theta_{\text{MLP}}, \psi_{\text{MLP}}$ is optimized with the TD loss as in Equations 4 and 5.

$$J_Q(\theta_{\text{MLP}}) = \mathbb{E}_{s,a,r,s' \sim \mathcal{D}}[(Q_{\theta_{\text{MLP}}}(f_{\theta_{\text{VLM}}}(s, a)) - r - \gamma V_{\bar{\psi}_{\text{MLP}}}(f_{\bar{\psi}_{\text{VLM}}}(s')))^2]. \quad (4)$$

$$J_V(\psi_{\text{MLP}}) = \mathbb{E}_{s \sim \mathcal{D}}\left[\mathbb{E}_{a' \sim \pi_\phi(\cdot|s)}[(V_{\psi_{\text{MLP}}}(f_{\psi_{\text{VLM}}}(s)) - Q_{\bar{\theta}_{\text{MLP}}}(f_{\bar{\theta}_{\text{VLM}}}(s, a')))^2]\right]. \quad (5)$$

### 4.2 BEST-OF-N POLICY EXTRACTION AGAINST THE LEARNED Q-FUNCTION

Given a learned Q-function, we will now use it to extract a policy in an efficient and reliable manner. Perhaps the most straightforward approach for doing so is to use the REINFORCE policy gradient

estimator to train the learned policy, however, this approach can be brittle with off-policy data. The presence of a "negative gradient" term Tajwar et al. (2024) (i.e., a term where the policy gradient multiplies the log likelihood by a negative-valued advantage) means that careful tuning of learning rates and interleaving policy and critic updates must be done to attain good performance (see Zhou et al. (2024b) Section 5.7 for a discussion of these challenges). While advantage-weighted supervised learning (i.e., AWR (Peng et al., 2019)) avoids this instability issue, it can be quite conservative in terms of moving away from the data collection policy.

To build a stable yet non-conservative policy training method, Digi-Q modifies weighted regression to make it more "aggressive", by leveraging the insight that access to a model of the Q-function allows for estimating values for multiple $N$ actions at any given state. After computing Q-values for multiple action candidates, we can imitate the best action. This would produce updates that are substantially less conservative than single-action AWR, without needing a negative gradient, and only employing a relatively more stable supervised learning update. Theoretically, this is because the implicit KL constraint against the data-generating policy that makes AWR conservative, is now much less of a problem with our multiple-action approach, since this implicit constraint is enforced against the Best-of-N policy (Cobbe et al., 2021), which already is much less conservative than the data collection policy for larger values of $N$. Moreover, akin to how verifiers have been used in reasoning problems, the value function critic can be used to score multiple possible actions in an offline manner without actually running the rollout or seeking a downstream correctness signal. This approach is similar to concurrent work Mark et al. (2024) from the domain of robotic policy learning.

Concretely, given any state $s$, we sample $N$ action token sequences from the initial policy: $a_1, \cdots, a_N \sim \pi_\beta(\cdot|s)$, where $\pi_\beta$ is the policy used to rollout the offline dataset. Now, we rank these actions according to the Q-values obtained from the value function trained previously. The policy is then trained to imitate the highest Q-value action of these $N$ actions as long as this best action also attains a positive advantage. Intuitively, this serves as a proxy learning objective to maximize the advantage function per state without the risk of "negative gradient". Formally, this means that the policy is optimized as per the loss described in Equation 6.

$$J_\pi(\phi) = \mathbb{E}_{s_t \sim \mathcal{D}, a_i \sim \pi_\beta(\cdot|s_t)} \left[ \sum_{i=1}^{N} \delta(a_i) \sum_{h=1}^{L} \log \pi_\beta(a_i^h|s_t, a_i^{1:h-1}) \right], \tag{6}$$

where $\delta(a_i) = \mathbb{1}\{a_i = \arg\max_i Q(s_t, a_i) \text{ and } Q(s_t, a_i) - V(s_t) > 0\}$. This approach allows us to make fairly non-conservative policy updates, while also being stable and efficient due to a log loss.

## 4.3 PUTTING IT TOGETHER: IMPLEMENTATION DETAILS

A pseudocode of our overall algorithm is shown in Appendix A. After initially fine-tuning the VLM through the representation fine-tuning scheme in Section 4.1, Digi-Q trains Q and V-functions before performing gradient updates on the actor, where the VLM backbone for the V-function is kept frozen from the pre-trained checkpoint. The usage of V-functions follows from Zhou et al. (2024b); Snell et al. (2023) to improve training stability. The actor is represented on top of a separate VLM and is trained end-to-end, unlike the use of frozen features for the critic to improve training stability of TD-learning. For our experiments, we sample $N = 16$ actions for computing the Best-of-N reranking policy learning objective in Equation 6: while the choice of $N$ can differ from domain to domain, our runs show that $N = 16$ is a good choice for our domain of Android device control. We use LLaVa-1.5 (Liu et al., 2024a) for the backbone VLM for our Q- and V- functions. The architecture details can be found in Appendix B.2.

## 5 EXPERIMENTAL EVALUATION

The goal of our experiments is to evaluate the efficacy of Digi-Q in producing effective Q-functions that in turn are able to train strong Android device control agents. Our experiments will answer the following questions: **(1)** How does Digi-Q compare with other state-of-the-art agent training algorithms, previously studied in the context of Android device control tasks? and **(2)** Can Digi-Q learn effectively from past interaction data? In addition, we perform several ablation experiments to understand the effects of various components of Digi-Q: to understand the benefits of using representation fine-tuning and to validate the efficacy of the Best-of-N reranking approach for training the policy using the value function.

| | | | AitW General | | AitW Web Shopping | |
|---|---|---|---|---|---|---|
| | | | Train | Test | Train | Test |
| **Prompting** | SET-OF-MARKS | GPT-4V | 5.2 | 13.5 | 3.1 | 8.3 |
| | | Gemini 1.5 Pro | 32.3 | 16.7 | 6.3 | 11.5 |
| | APPAGENT | GPT-4V | 13.5 | 17.7 | 12.5 | 8.3 |
| | | Gemini 1.5 Pro | 14.6 | 16.7 | 5.2 | 8.3 |
| **Learning** | SUPERVISED TRAINING | CogAgent | 25.0 | 25.0 | 31.3 | 38.5 |
| | | AutoUI | 27.7 | 22.9 | 20.7 | 25.0 |
| | OFFLINE | Filtered BC | $51.0 \pm 0.9$ | $54.5 \pm 1.3$ | $37.2 \pm 4.7$ | $43.8 \pm 1.7$ |
| | | DigiRL | $53.5 \pm 2.7$ | $59.0 \pm 4.7$ | $43.1 \pm 3.6$ | $47.6 \pm 4.2$ |
| | | **Digi-Q (Ours)** | $\mathbf{61.5 \pm 2.3}$ | $\mathbf{71.2 \pm 2.1}$ | $\mathbf{53.1 \pm 1.7}$ | $\mathbf{58.0 \pm 2.1}$ |
| | ONLINE | DigiRL | $63.5 \pm 3.1$ | $74.5 \pm 2.6$ | $52.6 \pm 1.6$ | $57.3 \pm 3.1$ |

Table 1: **Main comparisons of different agents across various settings.** Each offline experiment is repeated three times and the mean and standard deviation are reported. To be consistent with prior work (Bai et al., 2024), results are evaluated with the autonomous evaluator with the first 96 instructions in the train and test set.

## 5.1 MAIN PERFORMANCE RESULTS

**Comparisons.** We compare Digi-Q with prior methods for building Android device control agents. First, we compare Digi-Q with prompting-based methods that extend off-the-shelf proprietary VLMs such as GPT-4V (OpenAI, 2024b) and Gemini 1.5 Pro (GeminiTeam, 2024) with the Set-of-Marks (Yang et al., 2023) and a chain-of-thought mechanism for producing actions. We also compare with existing VLMs trained via imitation learning for device control: CogAgent (Hong et al., 2023), a 18B model. We keep results for these three approaches to be the same as scores previously reported in the DigiRL paper (evaluations were done in March 2024.[1]). That said, we evaluate AutoUI-1B (Zhang and Zhang, 2023) again. Finally, we compare to a state-of-the-art approach for training device control agents, DigiRL, which does not utilize a state-action Q-value function, but rather uses a state-only value function and MC return estimates to estimate advantages *only* on on-policy actions. We evaluate our results on Android-in-the-Wild (AitW) with offline dataset containing 1296 trajectories for AitW Web Shopping subset and 1008 trajectories from AitW General subset from pre-trained AutoUI checkpoint, following Bai et al. (2024). More details on the offline dataset can be found in Appendix B.3. To understand the ballpark of performance gains from Digi-Q, we also compare with DigiRL in the offline-to-online setting, which is given access to online interaction starting from a dataset of 512 initial trajectories.

**Results.** Our main results are presented in Table 1. We find that Digi-Q outperforms all prompting-based methods substantially (53.5% absolute improvement on average compared to the best prompting-based approach AppAgent with GPT-4V) and improves over the previous state-of-the-art in the offline setting, DigiRL by around 21.2% relatively averaged on General and Web Shopping test subsets, and 31.5% relatively over Filtered-BC, a simple but strong baseline. In fact, the performance of Digi-Q even roughly matches the performance of DigiRL when it is allowed to perform on-policy interaction. By visualizing the agent's rollouts on test examples, as we will show in Appendix C.1, we find that training a policy with value functions enhances the capability of RL to perform dynamic programming with sub-optimal data to learn a better policy in the environment.

## 5.2 ABLATION STUDIES

---

[1]We expect the latest evaluations of all methods to largely underestimate results from evaluations in March 2024 on the AitW Web Shopping subset because certain websites have started blocking agents from taking actions on the website. Details are shown in Appendix C.2. This issue affects both the baseline approaches and our method, and is a direct consequence of the non-stationarity and dynamism of the web environment. This also means that for a given fixed offline dataset, baseline performance numbers from March 2024 are expected to be higher than if the prior approach were run again as of the time of writing this paper. Hence if Digi-Q outperforms March 2024 evaluations of a prior method, we can reliably expect it to outperform that prior approach today. We also collected trajectory dataset with higher success rate for offline training, which we expect should lead to better results than March 2024, but also expect results to be uniformly underestimated due to the evaluation issue.

Next, we will perform a series of controlled experiments to understand the reasons behind the efficacy of Digi-Q. In particular, we will attempt to understand **(1)** the effect of representation fine-tuning (Stage I) for seeding the VLM representations for subsequent Q-function training, **(2)** the behavior of Best-of-N reranking style policy extraction operator compared to AWR (used by DigiRL (Bai et al., 2024)) and standard REINFORCE-style policy gradients (Williams, 1992), **(3)** the benefits of TD-learning over the more conventional approach of supervised regression to Monte-Carlo return for training the value function, and **(4)** the scaling performance with more data of Digi-Q compared to other baselines. Experimental details of the ablation studies can be found in Appendix B.4.

| Representation | Performance |
|---|---|
| Behavior Policy | 25.0 |
| Digi-Q (w/ MC return) | $37.5 \pm 4.5$ |
| Digi-Q Off-the-shelf VLM | $31.9 \pm 1.3$ |
| Digi-Q w/ BLIP-2 + BERT | $47.6 \pm 5.2$ |
| **Digi-Q (Ours)** | $\mathbf{58.0 \pm 2.1}$ |

Table 2: **Efficacy of our representation fine-tuning procedure** on the Web-Shopping test set in AitW.

**(1) The effect of representation fine-tuning in Digi-Q.** We first analyze the effect of fine-tuning the VLM representations by training them to accurately detect actions that led to a substantial change in the scene. To do so, we compare Digi-Q with alternate approaches that train Q-functions on top of two other natural choices of representations: **(a)** not using a generative VLM (i.e., Llava-1.5), but instead using frozen BLIP-2 (Radford et al., 2021) and tuned BERT (Devlin et al., 2019) representations, following Bai et al. (2024); **(b)** using an off-the-shelf VLM, without any representation fine-tuning (Chen et al., 2024).

| Actor Objective | Performance | KL |
|---|---|---|
| Behavior Policy | 25.0 | 0 |
| REINFORCE | $37.5 \pm 4.7$ | 7.15 |
| AWR | $19.4 \pm 1.3$ | 2.84 |
| Digi-Q | $\mathbf{58.0 \pm 2.1}$ | 3.28 |

Table 3: **(1) Performance and (2) token-level KL-divergence value between the learned policy and the dataset** when using different policy extraction methods on Web Shopping test set. We utilize the same critic for all the methods, and only train the policy differently.

As shown in Table 2, observe that simply using an off-the-shelf VLM only leads to marginal improvement over the behavior policy (31.9% compared to 25.0%): this is perhaps expected because an off-the-shelf generative VLM introduces a representation that captures features about the scene holistically, but does not necessarily judge whether these features are actionable. As we will also qualitatively show in Section C.1, off-the-shelf VLMs also do not pay enough attention to action information, resulting in a Q-function that degenerates to a similar solution as the state-only value function in the absence of aggressive action coverage in the data. In fact, Digi-Q using off-the-shelf VLM falls short of Digi-Q w/ BLIP-2 + BERT as well. In contrast, the representation fine-tuning procedure employed by Digi-Q is able to unlock the advantage of using rich VLMs and achieves more than 10% absolute improvement over the counterpart with BLIP-2 + BERT.

**(2) The effect of Best-of-N reranking style policy extraction operator.** Next, we aim to understand the impact of using Best-of-N reranking for policy extraction. This operator differs from traditional policy extraction methods in several ways: **(i)** the use of multiple actions **(ii)** not using a "negative gradient" (Tajwar et al., 2024) as in REINFORCE (Williams, 1992). To understand the effect of the number of actions in **(i)**, we ablate Digi-Q over multiple values of $N \in \{1, 4, 8, 16\}$ in Figure 3 (*Left*). Observe that Digi-Q improves monotonically as $N$ increases, indicating a clear benefit of sampling more actions and reranking them against the Q-function during training. More discussions on the ablations of number of actions resampled can be found in Appendix B.3.

Next, we answer **(ii)** by comparing Digi-Q with REINFORCE and supervised regression (AWR). Our results in Table 3 show that while REINFORCE is able to achieve some improvements compared to the data collection policy (37.5% compared to 25.0%), it also suffers from the highest variance among all ablated methods. We hypothesize that this is a direct consequence of the negative gradient, which is known to sometimes destabilize training. While AWR (Bai et al. (2024)) does not suffer from this issue, it is also not able to stably improve the policy (19.4% compared to 25.0%), likely because it is conservative. On the other hand, Digi-Q is able to make substantial improvements.

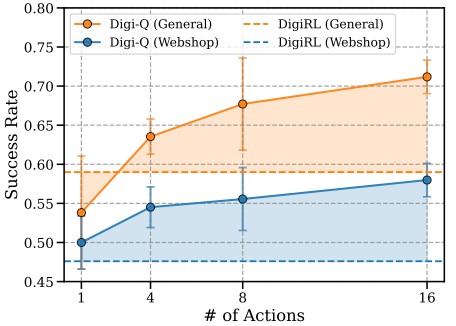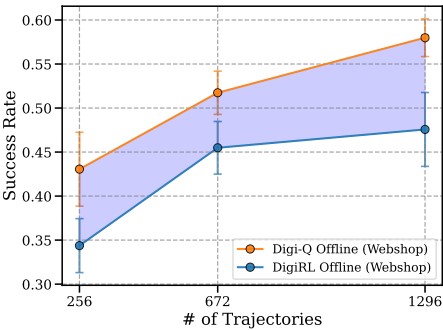

Figure 3: *Left:* **Performance of Digi-Q when varying the number of actions $N$ used for policy extraction.** Observe that the performance of Digi-Q improves when more actions are used for policy extraction, indicating the efficacy of our approach and the benefits of learning a Q-function. *Right:* **Data efficiency of Digi-Q and DigiRL.** The success rate of Digi-Q increases significantly faster than offline DigiRL given the same amount of more data.

Next we attempt to understand how "non-conservative" the updates made by different approaches are since one concern with AWR-style updates in prior work is the extent to which they are conservative. We wish to understand if our Best-of-N reranking based policy extraction approach also admits conservative updates. To do so, we measured the KL-divergence between actions from the dataset and the fine-tuned policies produced by Digi-Q, AWR, and REINFORCE in Table 3. Note that Digi-Q incurs a larger KL-divergence value unlike AWR that incurs the smallest deviation and is most conservative. In contrast, REINFORCE attains larger divergence values but behaves unstably (see Appendix C.3 for some example rollouts). Some qualitative examples for these variants are shown in Appendix C.1. Our results are also consistent with findings in concurrent work from robotic control problems (Mark et al., 2024).

**(3) The effect of TD-learning as opposed to MC.** To understand the importance of TD-learning for training the critic over Monte-Carlo (MC) regression that previous work is based on, we run an ablation of Digi-Q, which uses MC regression. Observe in Table 2, that this version underperforms Digi-Q by 20% (58.0% compared to 37.5%). As we show in Appendix C.1, value functions from MC regression exhibit high variance, which inhibits them from producing good policies even when used to rank multiple actions.

**(4) Scaling performance of Digi-Q with different amount of data.** We present the comparison between Digi-Q and DigiRL along the axis of the number of training trajectories in Figure 3 (*Right*). For a fair comparison, for Digi-Q we rerun all stages of training while varying the amount of training data. As shown in Figure 3 (*Right*), we observe that Digi-Q outperforms DigiRL in all regimes, even in the low-data regime with only 256 trajectories. We suspect this is due to the ability to reuse data and perform better per-step credit assignment, thanks to a reliable Q function.

## 6 CONCLUSION AND FUTURE WORK

We presented Digi-Q, an effective method for training VLM Q-value functions from offline data, specifically for training real-world device-control agents. At the core of our method is a representation fine-tuning procedure that induces actionable features from VLM useful for later TD-learning and a Best-of-N policy training method that makes the best use of the learned Q function from TD-learning. While we primarily focus on GUI agent tasks on Android devices, our methodology is general, compute efficient, and leads to substantial improvement in performance. We believe that these ideas and approach should transfer to new tasks as well and applying Digi-Q to new domains in an interesting avenue for future work. That said, using the critic in Digi-Q in an active online self-improvement loop will require a more sophisticated system design to speed up the experiment iterations and methods to robustify the critic as the distribution of the agent policy drifts far from the base data collection policy with more online improvement. Nonetheless, ideas from Kalashnikov et al. (2018) could provide a good starting point to build policy learning systems based on TD-learning during real-world interaction.

## ACKNOWLEDGEMENTS

Hao Bai would like to thank Nan Jiang, Shengbang Tong and Jiayi Pan for early discussions during the project. This work was partly done when Hao Bai was a visiting scholar at UC Berkeley. Aviral Kumar would like to thank Amrith Setlur for discussions. This work is supported by NSF IIS-2246811, ONR N00014-24-12206, and ONR N00014-21-1-2838. We thank Google Cloud for providing Gemini 1.5 Pro credit donations for academic use and some GPU and TPU resources. We also thank the NCSA Delta cluster admins for providing us with GPU resources for training.

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

# Appendices

## A  DETAILS ON THE ALGORITHM

For completeness, we include a detailed pseudo-code of Digi-Q in Algorithm 1. After initializing the parameters, we perform the representation fine-tuning procedure on top of VLM to obtain actionable features for later TD-learning. Then the VLM parameters will be kept frozen and we train the Q- and V- functions using TD-learning on top of frozen VLM representations. After both value functions are trained, we perform gradient updates on the actor with Best-of-N policy extraction.

---

**Algorithm 1** Digi-Q: Practical Framework

---

1: Initialize parameters $\phi, \psi_{\mathrm{MLP}}, \bar{\psi}_{\mathrm{MLP}}, \theta_{\mathrm{MLP}}, \bar{\theta}_{\mathrm{MLP}}$.
2: Initialize replay buffer $\mathcal{D}$ (from an offline dataset).
3: **for** each VLM iteration **do**
4:     $\theta_{\mathrm{VLM}} \leftarrow \nabla J_{\mathcal{P}}(\theta_{\mathrm{VLM}})$                        ▷ Equation 3
5: **end for**
6: **for** each critic step **do**
7:     ## Update high-level Q and V functions by target function bootstrapping.
8:     $\theta_{\mathrm{MLP}} \leftarrow \theta_{\mathrm{MLP}} - \nabla J_{\theta_{\mathrm{MLP}}}(Q)$            ▷ Equation 4
9:     $\psi_{\mathrm{MLP}} \leftarrow \psi_{\mathrm{MLP}} - \nabla J_{\psi_{\mathrm{MLP}}}(V)$             ▷ Equation 5
10:    ## Update target Q and V functions.
11:    $\bar{\theta}_{\mathrm{MLP}} \leftarrow (1 - \tau)\bar{\theta}_{\mathrm{MLP}} + \tau\theta_{\mathrm{MLP}}$
12:    $\bar{\psi}_{\mathrm{MLP}} \leftarrow (1 - \tau)\bar{\psi}_{\mathrm{MLP}} + \tau\psi_{\mathrm{MLP}}$
13: **end for**
14: ## Update low-level actor with high-level critic.
15: **for** each actor step **do**
16:    $\phi \leftarrow \phi - \nabla J_{\phi}(\pi)$                             ▷ Equation 6
17: **end for**

---

## B  EXPERIMENTAL DETAILS

### B.1  COMPUTE EFFICIENCY COMPARISON

A common concern with deploying TD-learning methods to train large-scale foundation models is their compute inefficiency (Abdulhai et al., 2023; Chebotar et al., 2023). Therefore, we attempted to understand the compute-performance tradeoffs associated with Digi-Q by comparing it against end-to-end TD-learning on VLMs without using any representation fine-tuning or frozen pre-trained representations. We plot the performance-compute tradeoff curve for Digi-Q on the web-shopping subset of the AitW dataset in Figure 4. We found it a bit hard to fine-tune an entire VLM with TD-learning, which required iteration on hyperparameters such as learning rate and soft update rates for target networks. Due to the compute-intensive nature, we use a 3B VLM (PaLiGemma (Beyer et al., 2024)) for these runs instead of our 7B VLM (Liu et al., 2024b), and evaluate the performance of the critic as measured by the correlation between advantage predictions and ground-truth notion of human judgement on a held-out set of trajectories. In particular, we find that end-to-end TD-learning exhibits a much worse performance-compute frontier, to the extent that beyond a point more training FLOPS hurts performance. We conjecture that this behavior is likely a result of well-known pathologies of training large models with TD learning (Kumar et al., 2022), though we leave it for future work to fully understand these pathologies in our context. In contrast, while Digi-Q invests an initial amount of computation for representation fine-tuning, its accuracy quickly rises up and results in much better frontiers, with no instability. The calculation of the FLOPS is shown below.

**FLOPS Calculation.** The 3B VLM takes $45.6 \times 10^{12}$ FLOPS for *each sample* for forward plus backward process. As the end-to-end TD learning contains one VLM as part of the Q function and one VLM as the target Q function (which only do forward pass), one sample takes $68.4 \times 10^{12}$ FLOPS (according to Hoffmann et al. (2022), the FLOPS incurred by the forward prrcess is approximately half of the backward process). Thus, as the longest run takes 15k samples, the last point of the

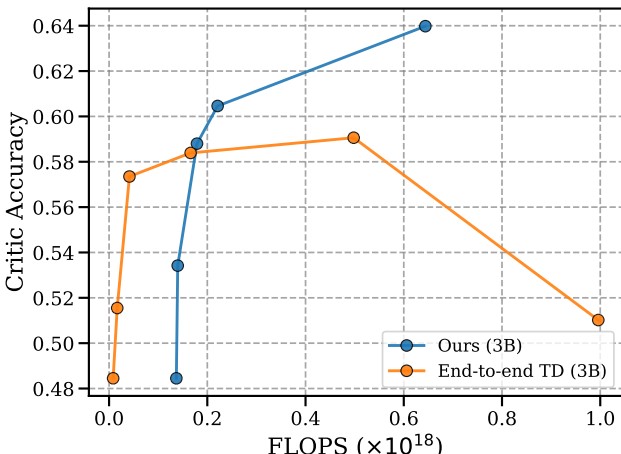

Figure 4: **Offline critic evaluation accuracy as a function of compute** measured in terms of training FLOPS, compared across Digi-Q, end-to-end TD-learning on a VLM, and MC return. Observe that the critic accuracy is much better for our approach over end-to-end TD-learning as the amount of compute increases.

end-to-end run in Figure 4 takes around $1 \times 10^{18}$ FLOPS. Also, the first logged point takes 128 samples, so the starting point should have $8.3 \times 10^{15}$ FLOPS.

On the other hand, in Digi-Q, we first finetune the 3B VLM, which incurs only one forward and backward process. Thus, finetuning the 3B VLM on 2000 samples takes $91.2 \times 10^{15}$ FLOPS. After that, we infer the representations of these samples with the 3B VLM, which includes one forward pass. This sums up to $136.8 \times 10^{15}$ FLOPs, which explains the starting point of the Digi-Q curve. Then we only train the value head using the VLM representations.[2] The size of the value head is 0.07B, incurring $1.1 \times 10^{12}$ FLOPS for each sample. The longest run of Digi-Q takes 0.46M samples, thus incurring $506.9 \times 10^{15}$ FLOPS ($10 \times 10^{17}$).

Thus, the end-to-end TD learning should range from $0.0083 \times 10^{15}$ to $1 \times 10^{18}$ FLOPS, while Digi-Q should range from $0.137 \times 10^{18}$ FLOPS to $0.644 \times 10^{18}$ FLOPS, which is shown in Figure 4.

**Critic Accuracy.** We manually label 483 states with binary advantages, and normalize the advantages produced by the agents to have a mean of zero before thresholding and calculating its accuracy with human annotations.

### B.2 CRITIC MODEL ARCHITECTURE

We show the details of the critic model architecture in Figure 5. In our environment setting, the states are composed of task, observation (screenshot at step $t$), previous observation (screenshot at step $t-1$), and previous action (action at timestep $t-1$). The task and previous action are text strings, while observations are images. We encode the text strings with BERT and images with BLIP-2 model. Then we concatenate all these feature vectors and pass them through a MLP that tries to predict the V value. The target of the V value is calculated by Equation 5.

The state-action features are modeled by the current action as well, which is a string passed into not only the BERT encoder but also a part of the prompt passed into the VLM. The prompt is described in Appendix D. In the end, the Q features include the BERT embeddings, the BLIP-2 embeddings, and the VLM intermediate-layer representations. We concatenate all of these feature vectors and pass into the another MLP that predicts the Q value. The target of Q value is calculated by Equation 4.

### B.3 TRAINING DATASET CONSTRUCTION

We use the pre-trained AutoUI checkpoint to collect offline trajectories. Specifically, to collect each trajectory, starting from the home screen, the agent generates an action, and then the environment takes the action and transitions to the next state. It iterates until a maximum number of steps have been reached or the autonomous evaluator has decided to be a success. We collect 1296 trajectories

---

[2]In this experiment, we fix the BERT model when running Digi-Q.

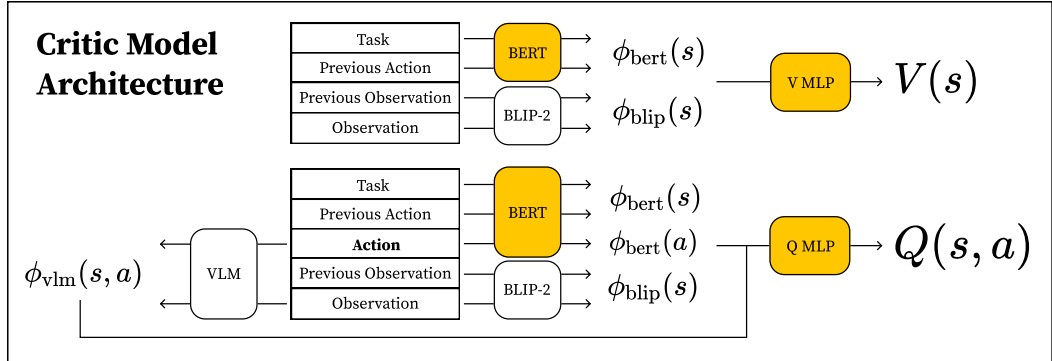

Figure 5: **Q-function architecture.** The modules marked orange are trained, otherwise the module is kept fixed.

this way for both AitW Webshop and AitW General subsets. The horizon $H$ of the Webshop subset is set to 20, and the horizon of the General subset is set to 10, which aligns with (Bai et al., 2024). Each trajectory is composed of state-action-reward-next-state pairs $(s, a, r, s')$, which is also referred to as "transitions".

The $N$ actions in the offline dataset used for Best-of-N loss are sampled post-hoc from the pre-trained AutoUI checkpoint. When training the actor offline, as we use the Best-of-N loss, we want to sample more than one action. From an engineering aspect, collecting actions each time we sample from the offline dataset $\mathcal{D}$ during training is not efficient. Thus, in practice, we pre-collect $K = 64$ actions for each state, and store them in the offline dataset. As $N \in \{1, 2, 4, 8, 16\}$ is much smaller than 64, this strategy serves as a good approximation and results in good performance. It suffices to give enough variety compared to sampling the actions when training the actor model. Note that in this case, the original action will always appear in the offline dataset.

### B.4 ADDITIONAL METHOD DETAILS

**Task set formulations.** The two task sets (Webshop and General) in the AitW dataset have different horizons $H$ (maximum number of steps allowed) in a trajectory to improve computational efficiency. Specifically, $H = 20$ for AitW Webshop and $H = 10$ for AitW General. Following tradition (Bai et al., 2024), we keep $A > 1/H$ (e.g. 0.05 for AitW Webshop) as a threshold for the actor model to learn the state-action pair.

**Ablation on representation fine-tuning and TD learning as opposed to MC.** In the ablation study on representation fine-tuning, for all configurations, we train the actor model with Best-of-N loss where $N = 16$ to keep computation efficient. This is also the case for the ablation on the TD learning as opposed to MC ablations.

**Ablation on actor loss.** For the ablation study on the actor loss, we keep the same trained Q function, while we ablate only on the loss used to train the actor model. We use 30 actor epochs for the Best-of-N loss and AWR loss, and 120 epochs for the REINFORCE loss as the magnitide of the raw advantage is very small. We use $N = 16$ for the Best-of-N loss, while REINFORCE and AWR both uses the original action in the offlin dataset.

**Value function**. In practice, we find the V function significantly easier to train, and it suffices to only use the representations of the state from the vision encoder of the VLM (CLIP) to train the V functions. This simplification significantly saves time and space required, and aligns with previous work (Bai et al., 2024).

## C QUALITATIVE EXAMPLES

### C.1 QUALITATIVE VISUALIZATIONS

**Qualitative comparisons between different value function learning approaches.** To qualitatively understand the quality of the Q-function learned, in Figure 6, we visualize advantage estimates

| Task | Go to newegg.com | | Go to walmart.com, search for 'alienware area 51' | | | Go to costco.com | | |
|---|---|---|---|---|---|---|---|---|
| $(s, a)$ | | | | | | | | |
| Human | -1 | 1 | -1 | 1 | 1 | -1 | 1 | 1 |
| **TD-Learning** | -0.03 | 0.09 | -0.26 | -0.12 | -0.01 | -0.27 | 0.06 | 0.06 |
| Monte-Carlo | 0.10 | 0.14 | 0.24 | 0.15 | 0.18 | 0.20 | 0.18 | 0.26 |
| TD (w/ CLIP+BERT) | -0.02 | -0.02 | 0.03 | 0.03 | 0.05 | -0.04 | -0.04 | 0.06 |
| TD (Off-the-Shelf VLM) | 0.04 | 0.03 | 0.06 | 0.07 | 0.20 | 0.02 | 0.05 | 0.32 |

Figure 6: **Qualitative examples showing the advantage estimations of several transitions of TD (ours), Monte-Carlo, and TD without VLM representation.** Advantage estimations using TD-learnt value functions top of VLM representation better align with human judgements compared to MC and TD without using VLM.

$A(s, a) = Q_\theta(s, a) - V_\phi(s)$ computed from Q-functions produced by four methods: **(1)** Digi-Q (with representation fine-tuning and TD-learning), **(2)** Monte-Carlo regression, **(3)** Digi-Q but using BLIP-2 + BERT representations from Bai et al. (2024); and **(4)** Digi-Q without representation fine-tuning. We compare advantages with human judgments of whether the actions mark progress towards the desired task. Ideally, good actions should attain a positive advantage. We observe that advantage estimates from MC regression are often erroneous and uncorrelated with the human notion of good actions, perhaps because of the use of high-variance MC estimator. Moreover, we find that **(3)** converges to a degenerate $Q(s, a)$ that approximately matches a state-only value function, with limited meaningful sensitivity to the action input, which is problematic for policy learning. **(4)** mitigates this problem and produces different action values under the same state, but still fails at fine-grained differences like clicking at different positions on the screen. Thus, all these ablation variants perform suboptimally at attaining a good correlation with human judgement, only **(1)** Digi-Q is able to produce advantage estimates that align well with human annotations and judgement.

## C.2 ENVIRONMENT ERRORS

We observe that several tasks has problems working in the environment introduced in Bai et al. (2024). We observe that (1) the `newegg.com` domain has a high probability of blocking the agent from accessing it, and (2) the `costco.com` domain prevents the agent from typing the `<ENTER>` key. Examples are shown in Figure 7. These problems were not observed in Bai et al. (2024). This is the main reason why some scores on the AitW Webshop subset in this paper falls a little behind Bai et al. (2024).

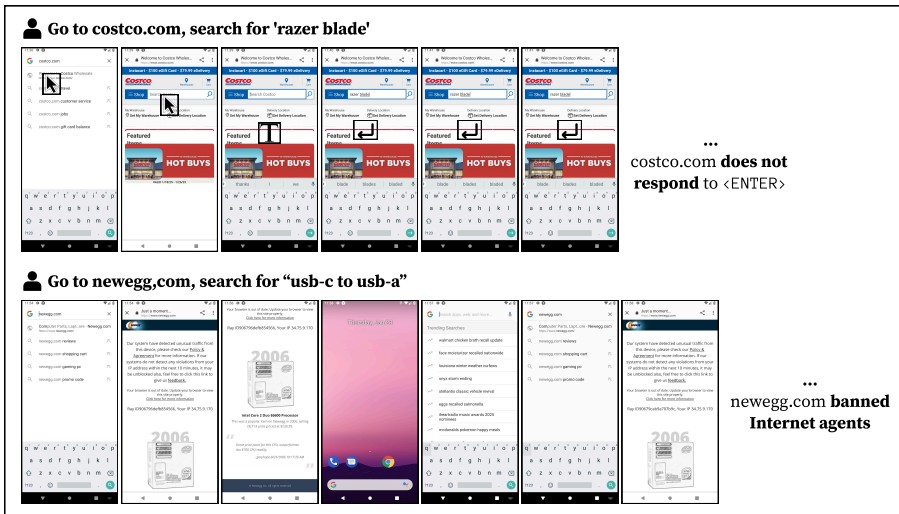

Figure 7: **Environment errors.** These errors are systematic and can not be removed by the agent.

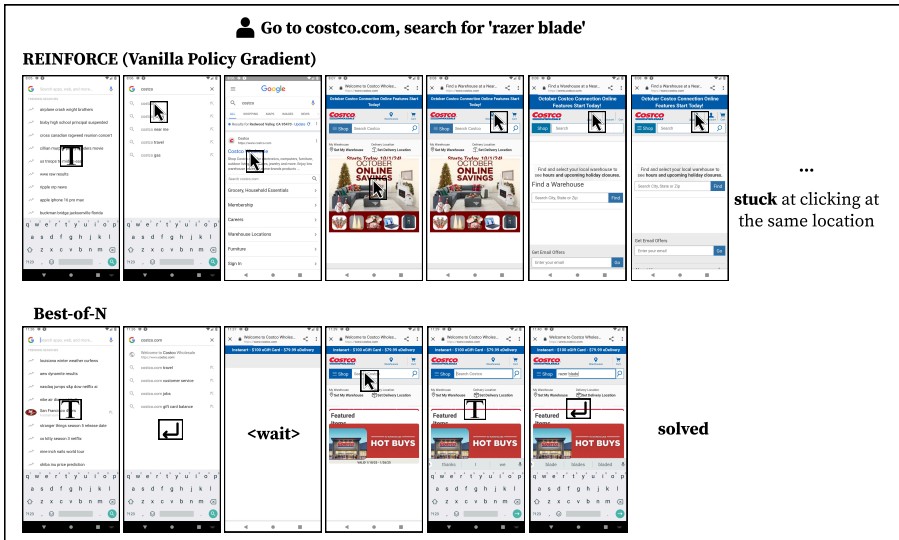

Figure 8: **Example trajectory of the agent trained with REINFORCE and Best-of-N loss.** Results show that the agent trained with REINFORCE tends to get stuck at a specific state because it's "stubborn", while agent trained with Best-of-N loss effectively solves the task.

Figure 9: **Trajectory examples showing benefits of Q-functions.** Our method can combine the best of a successful but lengthy (*A*) trajectory and a failed but short trajectory (*B*), to produce successful and short trajectories (*C*).

### C.3 EXAMPLE TRAJECTORY COMPARING REINFORCE AND BEST-OF-N LOSS

We show a typical trajectory produced by the agent trained with REINFORCE in Figure 8. We observe that the agent frequently diverges from the target and is too "stubborn" to recover from errors.

In this task towards searching for an item on costco.com, the agent has successfully arrived at costco.com, but (1) it takes some bad actions and (2) cannot recover. Specifically, after the agent clicks the warehouse button, it keeps clicking on the same button for 10 times until it clicks on somewhere else. This situation rarely appear in any trajectories collect by the agent trained with the Best-of-N loss.

### C.4 BENEFITS OF DYNAMIC PROGRAMMING

An appealing property of value-based RL is *dynamic programming*: the ability to identify optimal behaviors from overall suboptimal rollouts. We present a qualitative example in Figure 9 that illustrates this ability of Digi-Q in learning optimal behaviors from sub-optimal data. In this example, trajectory (A) and (B) are from the offline dataset where trajectory (A) successfully completes the task but has many redundant actions while trajectory (B) does not have redundant actions but fails to complete the task. It turns out that Digi-Q is able to learn a policy that performs dynamic programming with trajectory (A) and (B) to produce a trajectory (C) that completes the task in the most efficient way. Neither trajectory (A) nor (B) is the optimal trajectory for solving the task but this example shows the ability of Digi-Q to learn an optimal policy from sub-optimal data, which is theoretically impossible through imitation alone.

## D  VLM PROMPTS

The prompt we use for fine-tuning and inferring the VLM is shown in Figure 10. The prompt template is designed to be action-type-specific, in order to facilitate the VLM to better differentiate different types of actions, which promotes fine-grained representations within the same action type. The input to the VLM is constructed by the image and the text prompt. Note that the VLM only sees the current image (overlayed with a cursor if the action is to click), and the next image is only used to calculate whether the target should be "yes" or "no". The target is a single token to promote computational efficiency. In practice, we find that a long target sequence introduces challenges for the VLM to fine-tune the representations.

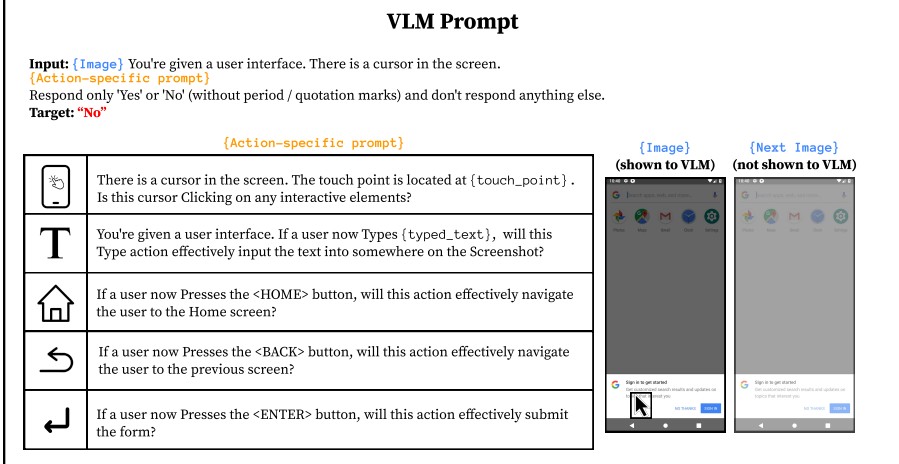

Figure 10: **Prompt template we use to fine-tune and infer the VLM.** The input prompt consists of an input image and text input. The text input include a template prompt concatenated with an action-specific prompt. The action-specific prompt includes specific information about the input image. The output (target) prompt is just a word "yes" or "no".

## E  HYPERPARAMETERS

Hyperparameters for Digi-Q are carefully tuned through binary search on the training set of General and Web Shopping subsets. The final choice of hyperparameters for both methods can be found in Table 4. Results for all other methods (Filtered BC and DigiRL) are kept the same as discussed in the original paper (Bai et al., 2024).

| Method | Hyperparameter | Value |
|---|---|---|
| Digi-Q | actor lr | 1e-4 |
| | value function lr | 1e-5 (general), 5e-6 (webshop) |
| | batch size | 128 |
| | maximum gradient norm | 0.01 |
| | actor updates per iteration | 30 |
| | value function updates per iteration | 20 |
| | number of iterations for offline actor updates | 15, 20, **30**, 45, 60 |
| | number of iterations for offline value function updates | 30, **40**, 45, 60, 90, 120 |
| VLM SFT | model checkpoint | liuhaotian/llava-v1.5-7b |
| | image aspect ratio | **pad**, no |
| | vision encoder | openai/clip-vit-large-patch14-336 |
| | number of training epochs | 3,**5**,8,10 |
| | per device train batch size | 8, **16**, 32 |
| | per device eval batch size | 4 |

Table 4: Hyperparameters for Digi-Q on both General and Web Shopping subset of AitW. If multiple values are displayed, the **bolded** value represents the selected value after hyperparamemter sweeping.

