# OpenReview forum: "Digi-Q: Learning VLM Q-Value Functions for Training Device-Control Agents"
_ICLR.cc/2025/Conference — ICLR 2025 Poster_

### Official Review · Reviewer_NJc5 · 2024-11-01

**Soundness:** 2
**Presentation:** 3
**Contribution:** 1
**Rating:** 3
**Confidence:** 5

**Summary:**

The paper presents Digi-Q, a value-based offline reinforcement learning (RL) approach aimed at training vision-language models (VLMs) for device control, specifically in Android GUI tasks. Digi-Q introduces a stable temporal-difference (TD) learning method on frozen VLM layers, optimizing Q-values while avoiding end-to-end model instability. Digi-Q also introduces a unique Best-of-N policy extraction that selects the best action among multiple candidates to improve policy performance without using traditional policy gradients.

**Strengths:**

1. The paper proposes an innovative Q-value-based RL approach, which integrates TD-learning with VLMs, to increase sample efficiency for complex environments.
2. The paper introduces Best-of-N policy extraction, enhancing policy learning stability by leveraging multiple action candidates.
3. The paper demonstrates improved compute efficiency over end-to-end TD learning, effectively addressing scalability in large models.
4. Problem formulations are okay, evaluations over the AiTW subset are comprehensive
5. Paper writing is good and presentations of results are good.

**Weaknesses:**

1. *Lack of Clear Motivation for Offline Value-Based Approach*: The paper does not sufficiently motivate the use of an offline, Q-value-based RL approach for device control, especially given the recognized stability and efficacy of methods like Advantage-Weighted Regression (AWR) and Generalized Advantage Estimation (GAE) as shown in previous work by [1] Bai et al. and [2] Pan et al. In particular, Q-value-based methods are known to introduce instability, especially in scenarios with partial observability, where AWR and GAE have demonstrated superior stability and simpler implementation when dealing with much more unstable and complex environments for on-device control.

2. *Limited Novelty Compared to DigiRL*: The paper's novelty is questionable when compared with previous works, especially DigiRL [1] by Bai et al. While Digi-Q proposes certain adaptations, such as the Best-of-N policy extraction, these contributions appear to be incremental rather than fundamentally advancing the state of value-based RL for device control.

3. *Concerns Over Experimental Data Reliability*: The experimental results lack reliability, particularly in light of my own testing experience. Many observed metrics and success rates in Digi-Q’s experiments suggest significant variance, casting doubt on the robustness of the results. Additional benchmarks and repeated trials would help validate these findings and ensure their reproducibility.

4. *Assumption of High-Quality Offline Data Set*: The paper's methodology hinges on a high-quality, well-curated offline dataset (e.g., AiTW), assuming this accurately represents all relevant scenarios. However, for real-world device control applications, where app behaviors and mobile environments change frequently, the method should ideally support a combination of offline and online data collection. Relying solely on offline data for pretraining limits adaptability, and the paper does not provide sufficient insight into how the pretrained policy can improve performance in dynamic online interaction settings.

5. *Lack of Evidence for Scalable Training at Scale*: Although the paper posits the question, “Can we train VLM agents at scale with value-based RL?” it falls short in demonstrating this scalability. There is a lack of empirical evidence supporting large-scale training or fine-tuning experiments, and the scalability of Digi-Q in practical, resource-intensive environments remains unclear without these demonstrations.

[1] Hao Bai, Yifei Zhou, Mert Cemri, Jiayi Pan, Alane Suhr, Sergey Levine, and Aviral Kumar. Digirl: Training in-the-wild device-control agents with autonomous reinforcement learning, 2024.

[2] Jiayi Pan, Yichi Zhang, Nicholas Tomlin, Yifei Zhou, Sergey Levine, and Alane Suhr. Autonomous evaluation and refinement of digital agents, 2024.

**Questions:**

1. In Table 1, the main comparisons of different agents across various settings raise significant concerns about the reliability and consistency of the reported results. The paper claims, "To be consistent with prior work (Bai et al., 2024), results are evaluated with the autonomous evaluator with the first 96 instructions in the train and test set." However, it appears that results for GPT-4V, Gemini 1.5 Pro, and CogAgent were directly copied from the DigiRL paper [1], while experiments were "reconducted" only for AutoUI and DigiRL. Notably, the results for AutoUI show a significant improvement compared to the previously reported figures in [1], while DigiRL’s offline results are selectively reduced.  This selective approach to data raises substantial concerns. If the intention was to reproduce results, it would be expected that any shifts in performance would be consistent across all models, not selectively applied. Furthermore, a success rate fluctuation of up to **5%** relative to previously reported results, given that the reported improvements are relatively modest, calls into question the robustness and reliability of the findings. Such fluctuations suggest that the experimental setup or evaluation may not be sufficiently stable, casting doubt on the paper's claims of improvement. I would appreciate clarification regarding the rationale for selectively re-evaluating some baselines and not others, as well as an explanation for the considerable performance variance observed. Without such transparency, the contributions of this work appear uncertain and potentially unreliable.

     **Hopefully, you can answer my concerns regarding these messy results or the "free lunch results" you used here.**

2. Why did you only evaluate your model on two subsets of AiTW? Could you explain the decision not to include other tasks, such as app installation, which would offer a broader evaluation of your model’s capabilities?

3. How similar are the evaluation tasks to those used during training? Please clarify the degree of overlap or differences, as this impacts how well the model generalizes beyond its training set.




[1] Hao Bai, Yifei Zhou, Mert Cemri, Jiayi Pan, Alane Suhr, Sergey Levine, and Aviral Kumar. Digirl: Training in-the-wild device-control agents with autonomous reinforcement learning, 2024.

---

> ### Author Response · Authors · 2024-11-20
> **Official Rebuttal to Reviewer NJc5 (1/2)**
>
> Thank you for your thorough review and feedback on our paper. We appreciate your constructive comments, which have helped us further clarify our motivations, methodology, and contributions. In response, we have made several updates to the paper, highlighted in blue, to address concerns regarding the novelty of our Q-value-based approach, the robustness of our results, and the scalability of our method.
>
> **Please let us know if these responses address your concerns and if so, we would be grateful if you would be willing to raise your score. We remain available for further discussion.** Below, we address your points in detail:
>
> > **Q-value-based methods are known to introduce instability, especially in scenarios with partial observability, where AWR and GAE have demonstrated superior stability and simpler implementation when dealing with much more unstable and complex environments for on-device control.**
>
>
> We agree that historically Q-function-based methods have been unstable, but we believe that an approach for making Q-value-based RL stable and feasible in a real-world problem of device control is of value and interest to the community, especially in light of our results which show that Digi-Q substantially outperforms DigiRL when trained from historically collected data.
>
> In regards to the motivation, prior algorithmic works in offline RL have shown the potential for Q-value based methods to be much more sample-efficient than purely AWR and GAE style methods (CQL [7] in traditional deep RL and ILQL[3] for language models). Such positive results contributed to our motivation for studying value-based RL in the device control domain to see if such an advantage of value-based RL still holds in this realistic setting.
>
>
> > **These contributions appear to be incremental rather than fundamentally advancing the state of value-based RL for device control.**
>
> To the best of our knowledge, we are not aware of any prior work in device control that utilizes a state-action Q-function $Q(s, a)$ for learning and attains state-of-the-art results in learning from static data, and therefore think that our contribution is of significance in terms of advancing the state-of-the-art of device control. Perhaps the closest work to us is DigiRL, but **note that this prior work does not train a Q-function at all and uses no Bellman backup**. In the offline stage, it simply trains a state-only value function by regressing against Monte-Carlo return estimates. Training a Q-function is more challenging since the VLM has to learn to relate pixel-based action coordinates on a screen with the image itself (and hence it requires several important algorithmic designs), but also leads to better results.
>
> If we were to directly follow the design of Bai et al. (NeurIPS 2024) to train the Q-function, as we already show in Table 2 (“Digi-Q w/ CLIP + BERT” row), this does not show much improvement compared to the behavior policy. Naively using TD learning to fine-tune the entire VLM does not work either due to the instability of TD learning as shown in Figure 3 (left).
>
> Once we have a Q function, we can optimize our policy with the Q function through sampling the actions and evaluating with Q function, opening up new possibilities of more efficient policy extraction methods that are infeasible with DigiRL. **It is not at all clear how one would apply Best-of-N policy extraction to DigiRL.** All of these differences result in superior performance for Digi-Q. Given these differences, improvements in performance, and the first state-of-the-art result showing value-based RL in device-control, we think our paper should be of significance.
>
> > **Many observed metrics and success rates in Digi-Q’s experiments suggest significant variance, casting doubt on the robustness of the results.**
>
> Although the metric of average over 3 seeds introduces a standard deviation of 2%, using 3 seeds is a compromise given the practical constraints and consistent with prior works in the device control domain (Table 1 in DigiRL). We would like to note that evaluations in the device control domain are much more costly and slow compared to experiments on standard deep RL benchmarks such as MuJoCo and Atari. We follow the DigiRL setting, where each evaluation involves 96 times of restarting and controlling a real Android emulator and can take more than 6 hours (more than 300 times slower than interactions on MuJoCo and Atari) on a T4 machine that we are using. The evaluation is also expensive, as queries into Gemini-1.5-Pro takes around $10 for evaluating every 100 trajectories. Additionally, the size of our 7B network is more than 1000 times larger than typical 3-layer convolutional neural networks used in MuJoCo and Atari (with fewer than 7M parameters). We are working on obtaining more compute and Gemini credits so that we will include the results of five seeds in the final version.
>
> (1/2)

---

> ### Author Response · Authors · 2024-11-20
> **Official Rebuttal to Reviewer NJc5 (2/2)**
>
> > **Assumption of High-Quality Offline Data Set: the paper's methodology hinges on a high-quality, well-curated offline dataset (e.g., AiTW), assuming this accurately represents all relevant scenarios.**
>
> _We think there might be some misunderstanding here, that **AitW is a task set**_ (i.e., it only prescribes a set of prompts / instructions), not an offline dataset of trajectories. The offline dataset is collected using a pre-trained initial policy, AutoUI, which only has around 20% success rate, similar to the protocol in DigiRL. So the offline data is far from high-quality and well-curated. The data collection step is exactly the same as [1], thus the offline data is not intentionally curated.
>
> > **There is a lack of empirical evidence supporting large-scale training or fine-tuning experiments, and the scalability of Digi-Q in practical, resource-intensive environments remains unclear without these demonstrations.**
>
> To the best of our knowledge, training 7B vision-language model Q-functions represents one of the largest scale experiments using TD-learning to date, benefiting from the idea of a separate representation fine-tuning. While this may not be the largest scale in industry, we are unaware of any published work that trains critics of this size. Note that we are already a magnitude larger than 200M critics used several prior works [2, 3, 4], that have already been published. That said, we are happy to tone down this claim or remove it altogether if you think that would be beneficial.
>
> > **I would appreciate clarification regarding the rationale for selectively re-evaluating some baselines and not others, as well as an explanation for the considerable performance variance observed.**
>
> We reproduced the strongest baselines due to compute and budget constraints during the submission process. This is because each evaluation involves 96 times of restarting and controlling a real Android emulator and can take more than 6 hours (more than 300 times slower interaction) on a T4 GPU that we are using and costly queries into Gemini-1.5-Pro (around $1 for every 10 trajectories). This reproduction is necessary due to the non-stationary nature of device control problems.
>
> To address these challenges and ensure a fair comparison, we re-collected offline data following the original procedures outlined for DigiRL but using up-to-date software and webpages. This re-evaluation yielded an improved performance for DigiRL (averaged 49.8% across task slices) compared to the originally reported results (averaged 48.7% across task slices). Thus results of our comparisons are relatively stable.
>
> The observed performance variance, particularly a fluctuation of up to 5% on AutoUI relative to previously reported results, reflects the challenges of working with real-world, non-stationary environments in device control. However the results for DigiRL hints at the fact that perhaps this non-stationarity is lower for RL methods. This challenge was discussed in Figure 4 of the DigiRL paper.
>
>
> > **Could you explain the decision not to include other tasks, such as app installation, which would offer a broader evaluation of your model’s capabilities?**
>
> We clarify that we use the same task set as prior work, DigiRL. While these tasks would indeed be broader, as described in [1], other tasks are either not suitable for scientific projects (e.g. tasks that involve accounts logging in) or have a very slow response time (e.g. app installation). We have added this discussion in Section 6 and noted this as a limitation of our work, which is no different from past work in this area.
>
> > **How similar are the evaluation tasks to those used during training? Please clarify the degree of overlap or differences, as this impacts how well the model generalizes beyond its training set.**
>
> We would like to note the task split is kept the same as mentioned in DigiRL [1], which in turn follows the standard in the device control community [5, 6]. While the current capability of device control models prevents them from generalizing to tasks too different from the training sets, this is not relevant to the main contribution of DigiQ which focuses on more efficient RL training algorithms.
>
> [1] Bai, Hao, et al. "DigiRL: Training In-The-Wild Device-Control Agents with Autonomous Reinforcement Learning." 2024.
>
> [2] Zhou, Yifei, et al. "ArCHer: Training Language Model Agents via Hierarchical Multi-Turn RL." 2024.
>
> [3] Snell, Charlie, et al. "Offline RL for Natural Language Generation with Implicit Language Q Learning." 2023.
>
> [4] Hong, Joey, et al. "Zero-Shot Goal-Directed Dialogue via Reinforcement Learning on Imagined Conversations." 2022.
>
> [5] Rawles, Christopher, et al. "Android in the Wild: A Large-Scale Dataset for Android Device Control." 2023.
>
> [6] Hong, Wenyi, et al. "CogAgent: A Visual Language Model for GUI Agents." 2024.
>
> [7] Kumar, Aviral, et al. "Conservative Q-Learning for Offline Reinforcement Learning." 2020.
>
> (2/2)

---

> ### Comment · Reviewer_NJc5 · 2024-11-25
> **Official Comment by Reviewer NJc5**
>
> ## A. Strong Critique of Experimental Results and Baseline Reproduction
> I must express my significant concerns regarding the integrity of your experimental results and the manner in which you have handled baseline reproductions. Your claim that “We **reproduced** the strongest baselines due to computing and budget constraints during the submission process” is highly questionable. The baseline numbers in your initial rows almost identically mirror those from previous studies, suggesting that you may have **copied** these results rather than genuinely reproducing them. This **selective** copying undermines the trustworthiness of your entire experimental framework and raises serious doubts about the authenticity of your findings.
> Furthermore, the discrepancies in your AutoUI results compared to DigiRL are glaring and unacceptable. Reporting an AutoUI performance of 27.7 against DigiRL paper’s 12.5 without a clear, consistent methodology indicates that you are comparing results from fundamentally different experimental setups. This lack of consistency not only skews the comparisons but also severely damages the credibility of your work. Your justification for attributing these differences to non-stationary environments is insufficient and does not adequately explain the substantial variances observed. If budget and compute constraints prevented you from conducting thorough and consistent experiments, you should have been transparent about which results were sourced from previous work or omitted them entirely to maintain the integrity of your study.
>
>
> ## B. Additional Critique on Generalization Abilities to Other Tasks in AiTW
> Furthermore, I am deeply concerned by your decision to exclude other tasks, such as app installation, from your evaluation. This omission severely limits the assessment of your model’s true capabilities and generalization potential. Based on my personal evaluation experiments with DigiRL, it is evident that models trained on AiTW general and web shopping tasks struggle significantly with generalization, often failing to perform even adequately on seemingly straightforward tasks. Given that your work is closely aligned with DigiRL, the absence of a broader range of tasks in your evaluation raises serious doubts about the robustness and versatility of your model. Without demonstrating performance across a diverse set of tasks, it is impossible to ascertain whether your approach truly advances the field or merely performs well within a narrow scope.
> Moreover, your lack of discussion or efforts to address generalization issues is particularly troubling. Considering the well-documented challenges faced by similar models in adapting to varied environments, it is imperative that you provide a comprehensive analysis of how your model fares beyond the narrowly defined tasks presented. Ignoring this critical aspect not only undermines the credibility of your work but also leaves significant gaps in understanding its practical applicability. I strongly expect to see including evaluations on additional tasks and to engage in a transparent discussion about the generalization capabilities of your model.
>
> ## Other comments
> In your discussion of training the Q-function, you state: “If we were to directly follow the design of Bai et al. (NeurIPS 2024) to train the Q-function, as we already show in Table 2 (‘Digi-Q w/ CLIP + BERT’ row), this does not show much improvement compared to the behavior policy. Naively using TD learning to fine-tune the entire VLM does not work either due to the instability of TD learning as shown in Figure 3 (left).”
> However, my thorough review of the DigiRL codebase and their publication reveals that they employ BLIP instead of a straightforward combination of CLIP and BERT. BLIP is a distinct model architecture that, while sharing some underlying principles with CLIP and BERT, incorporates unique components and training strategies. You need to be "accurate" in any case or at least explain the confusing concepts here.
>
> ## Summary of Concerns
> In summary, your experimental methodology is marred by questionable reproduction of baselines, significant discrepancies in key results, and a glaring lack of evaluation on diverse tasks necessary for demonstrating true generalization. These issues collectively render your findings unreliable and your contributions questionable. Also, I find the contributions of your work to be only marginal when compared with DigiRL. While DigiRL has established a robust framework with comprehensive evaluations and demonstrated significant advancements in the field, your work falls short in offering substantial improvements or novel insights.
>
> **I will firmly keep the score as “Reject”**

---

> > ### Author Response · Authors · 2024-12-01
> > **Follow-up discussion towards Reviewer NJc5**
> >
> > Thanks a lot for getting back to us! To address the concerns, we have now rerun several baselines, updated the paper with the new numbers, and are running the remainder. In summary, we find that all baselines perform similarly to the numbers in the DigiRL paper. For example, on the AitW General set, there is only a ~2% difference in success rate for the Set-of-Marks (Gemini-1.5-Pro) tasks, and ~5% difference in success rate for the CogAgent tasks performing worse than what is reported in the DigiRL paper. That said, please do note that baseline performance for prompting-based methods is expected to vary from time to time, as proprietary model checkpoints keep on evolving in addition to the non-stationarity of the task itself. Moreover it costs us $1000 to run one single evaluation extensively, which is why we did not run them earlier, but now we are adding these methods for each of our tables. We want to reiterate that our intention here in the paper was not to hide numbers or selectively rerun baselines, we simply made a logical compromise to run the most promising baseline (which involved training) as opposed to prompting-based methods or methods based on off-the-shelf models in the submission. Since our latest numbers for these methods are largely worse, this implies that DigiQ is still the most performant method and no conclusions change.
> >
> > We also clarify that we study exactly the same set of tasks as DigiRL, and choose to not study tasks like app installation for the same reasons as the emulator environment of DigiRL discards them: app installation tasks causes security reasons because an account is needed, and the Single subset fails to examine multi-step challenges that we’re interested in. While we agree that adding these tasks is important for future work and we will note this in the paper, we believe that a fair comparison on all tasks that our most related prior work studies should not be grounds for rejection. We clarify more on this below as well.
> >
> > > Strong Critique of Experimental Results and Baseline Reproduction
> >
> > We have reproduced some baseline results based on the DigiRL paper, and we have updated them in Table 1 of the paper in blue. The updates are copied below:
> >
> > | Method                        | AitW General (Train) | AitW General (Test) |
> > | ----------------------------- | -------------------- | ------------------- |
> > | Set-of-Marks (Gemini-1.5-pro) | 32.3/**30.2**        | 16.7/**14.6**       |
> > | CogAgent                      | 25.0/**18.8**        | 25.0/**29.5**       |
> >
> > *Bolded numbers are results we we-ran during the rebuttal period. Unbolded numbers are original results from the DigiRL paper.*
> >
> > These experiments are run under our own emulation environment, so the scores can be directly comparable to what we get from AutoUI/offline RL results. From the reproduction results we can see that the scores are more or less around the performance of DigiRL. Note that the original DigiRL paper also ran these experiments with only one run (there’s no standard error for these experiments in the table), so a reasonable variance is expected. The reason that we chose Gemini 1.5 Pro to reproduce instead of GPT-4V is that the original GPT-4V model API has been removed by OpenAI, but the `gemini-1.5-pro` model is still there.
> >
> > We want to kindly note that we’re still running more baseline experiments under our own environment. We hope to carry out most baseline results before the end of the rebuttal phase. We will include these updated results in later revisions of the paper.
> >
> > > Additional Critique on Generalization Abilities to Other Tasks in AiTW
> >
> > Our experiment setup and evaluation tasks are identical to DigiRL, because the focus of this work is to develop a better RL algorithm for device control instead of a generalist model checkpoint. Thus, the reason that we don’t include these tasks is the same as DigiRL (see Appendix A.1 Paragraph 1 in the DigiRL ArXiv paper):
> >
> > “ *The Android in the Wild (AiTW) task set is a large-scale dataset for android device control, containing five subsets: GoogleApps, Install, Web Shopping, General, and Single, where we select the General and Web Shopping subsets. Single subset is not considered here because all tasks in Single can be completed within one step and thus this subset fails to examine the multi-step challenges that we are interested in this paper. Install and GoogleApps are not considered due to security reasons as those tasks require an active Google account and parallel emulations can flag security concerns.*”
> >
> > > You need to be "accurate" in any case or at least explain the confusing concepts here.
> >
> > We apologize that we made the description vague and imprecise. In the context of “CLIP + BERT”, what we really meant is “BLIP + BERT”. We have updated this in the later revisions of the paper. Our experiments are based on the original DigiRL codebase, so the image encoder was kept the same.

---

### Official Review · Reviewer_3SUU · 2024-11-03

**Soundness:** 3
**Presentation:** 3
**Contribution:** 2
**Rating:** 6
**Confidence:** 3

**Summary:**

This paper introduces Digi-Q, a novel approach to making reinforcement learning work with large vision-language models (VLMs) for device control tasks. The authors tackle a challenging problem: while value-based reinforcement learning methods like Q-learning are known to be efficient, they've been notoriously difficult to use with large language models. The key insight of this work is that instead of trying to train the entire VLM using temporal difference (TD) learning, they first fine-tune the model's internal representations to better capture action-relevant features, then freeze these representations and only train a small Q-function on top. They also introduce a "Best-of-N" policy extraction method that samples multiple potential actions and trains the policy to imitate the ones rated highest by the Q-function. The authors evaluate their approach on Android device control tasks, showing improvements over previous methods and better computational efficiency than end-to-end TD learning. While the improvements are modest (about 10% better than previous methods) and limited to one domain, the work presents a practical approach to combining value-based reinforcement learning with large vision-language models, supported by thorough empirical validation and ablation studies.

**Strengths:**

The paper demonstrates several notable strengths across different dimensions. On the technical side, it successfully adapts Q-learning to work with large VLMs in a practical way. The two-phase approach of fine-tuning representations before freezing them for Q-learning is clever and addresses real computational challenges, while the Best-of-N policy extraction method offers a more stable alternative to traditional policy gradients (though the improvements are modest). The empirical work is thorough, with comprehensive ablation studies and comparisons against strong baselines like GPT-4V and Gemini, backed by proper statistical reporting across multiple runs. The presentation is clear and well-structured, with effective use of figures and helpful qualitative examples that illustrate how the method works in practice. From a practical perspective, the work addresses a real problem that practitioners face when trying to use Q-learning with large models, and while the 9.9% improvement isn't revolutionary, it represents meaningful progress. Importantly, the authors provide complete implementation details and hyperparameter choices, making their work reproducible. While none of these strengths are groundbreaking on their own, together they represent a solid engineering advance that makes value-based RL more practical with large models. The work is particularly strong in its empirical validation and clarity of presentation, even if the core technical innovations are relatively straightforward extensions of existing ideas.

**Weaknesses:**

The paper has several notable limitations that temper its impact. Most significantly, the evaluation is restricted to a single domain (Android device control), making it unclear whether the approach generalizes to other types of agent tasks or VLM applications. While the authors show a 9.9% improvement over previous methods, this is a relatively modest gain that comes with considerable complexity in the training pipeline. The theoretical foundation for the Best-of-N policy extraction approach is somewhat thin - while it works empirically, we lack a clear understanding of why this particular method is effective or how to choose the optimal value of N. The computational efficiency claims, while promising, would benefit from more detailed comparisons across different model scales and task complexities. There are also some concerning gaps in the analysis: the authors don't thoroughly explore failure cases or limitations of their method, and the stability analysis across different random seeds and hyperparameters could be more comprehensive. From a technical perspective, while the idea of fine-tuning representations before freezing them for Q-learning is practical, it's a relatively straightforward combination of existing techniques rather than a fundamental advance in how we approach VLM training. The ablation studies, while thorough in some areas, don't fully explore the sensitivity of the method to various design choices, particularly in the representation fine-tuning phase. Finally, the paper would benefit from a more detailed discussion of the computational resources required for training, as this is crucial information for practitioners considering adopting this approach.

**Questions:**

1. Could you discuss whether and how this approach might generalize to other domains? Have you attempted any preliminary experiments with different types of agent tasks?
2. The Best-of-N policy extraction method lacks strong theoretical justification. Could you provide more insight into why this approach works better than alternatives? How did you choose N=16 as the optimal value, and how sensitive is the method to this choice?
3. While you show improved compute efficiency compared to end-to-end TD learning, could you provide more concrete details about the total computational resources required for training? This would help practitioners better understand the real-world applicability.
4. Could you provide more detailed analysis of training stability across different random seeds and hyperparameters? The current results show standard deviations, but a deeper analysis would be valuable.
5. Could you provide examples of scenarios where your method struggles and analyze why these failures occur? The paper would benefit from a more thorough discussion of failure cases.
6. How dependent is your method on the specific VLM architecture used? Have you tested with different VLM backbones, and if so, how does the performance vary?
7. The representation fine-tuning phase seems crucial to your method's success. Could you provide more details about how sensitive the method is to different fine-tuning objectives or architectures? Have you explored alternative approaches to making VLM representations more action-aware?

---

> ### Author Response · Authors · 2024-11-20
> **Official Rebuttal to Reviewer 3SUU (1/3)**
>
> Thank you for your thorough review and constructive feedback on our paper. To address your concerns, we have updated the manuscript with revisions highlighted in blue to improve clarity, precision, and transparency regarding our methodology, scope, and experimental setup. Specifically, we clarified our focus on demonstrating the improved sample efficiency of TD-learning in the realistic setting of device control, conducted additional experiments to explore sensitivity and failure cases, and provided detailed insights into computational efficiency and theoretical underpinnings. Below, we respond to each of your comments in detail, incorporating new results, clarifications, and updates to the paper.
>
> **Please let us know if these responses address your concerns and if so, we would be grateful if you would be willing to raise your score. We remain available for further discussion.** Below, we address your points in detail:
>
> > **Most significantly, the evaluation is restricted to a single domain (Android device control), making it unclear whether the approach generalizes to other types of agent tasks or VLM applications.**
>
> We would like to clarify that our intention was to particularly show the efficacy of value-based RL in the scope of Android device-control settings, and not for general VLM agent problems. We have updated the wording in the paper to remove any phrases that might have given an impression otherwise.
>
> With regards to this problem setting, we believe that device control is already more general than several problem domains (e.g., shopping, travel planning, etc) that have been considered individually. In fact, our work is already in a more general setting than work in foundation agents that appears in ICML / NeurIPS / ICLR [1, 2, 3, 4]. Additionally, we use two subsets of AitW that focus on different parts of device control with around 200 tasks each (web shopping, device management; see Tables 2, 3). Finally, our closest prior work DigiRL (Bai et al. NeurIPS 2024) also focuses on only the problem of Android device control, but it was deemed to be of value and significance by the NeurIPS community.
>
> [1] Yao, Shunyu, et al. "WebShop: Towards Scalable Real-World Web Interaction with Grounded Language Agents." Advances in Neural Information Processing Systems 35 (NeurIPS 2022): 20744–20757.
>
> [2] Deng, Xiang, et al. "Mind2Web: Towards a Generalist Agent for the Web." Advances in Neural Information Processing Systems 36 (NeurIPS 2023): 922–940.
>
> [3] Zheng, Boyuan, et al. "GPT-4V(Ision) Is a Generalist Web Agent, If Grounded." Proceedings of the 41st International Conference on Machine Learning (ICML 2024).
>
> [4] Gur, Izzeddin, et al. "TravelPlanner: A Benchmark for Real-World Planning with Language Agents." Proceedings of the 41st International Conference on Machine Learning (ICML 2024).
>
> > **The theoretical foundation for the Best-of-N policy extraction approach is somewhat thin - while it works empirically, we lack a clear understanding of why this particular method is effective or how to choose the optimal value of N.**
>
> Theoretically, Best-of-N policy extraction simply imitates the action with the highest advantage within the distribution of the policy. As suggested by the theory of Conservative Policy Iteration (CPI), if the new policy achieves a higher advantage in that $𝔼_{s \sim d_{\pi^t}} 𝔼_{a \sim \pi^{t+1}(\cdot|s)} A^{\pi_t}(s,a) > 𝔼_{s \sim d_{\pi^t}} 𝔼_{a \sim \pi^{t}(\cdot|s)} A^{\pi_t}(s,a) = 0$ and each step is conservative in that $\pi^{t+1}(\cdot|s)$ and $\pi^{t}(\cdot|s)$ are close, then it is guaranteed that $\pi^{t+1}$ can achieve better performance compared to $\pi^t$. This theoretical guarantee is similar to that of PPO and TRPO.
>
> For choosing the best N empirically, we have provided an ablation in Figure 3 (right) showing that the larger the N is the better when N<16 (as shown by the monotone increasing curve from $N=1$ to $N=16$). However, the marginal performance improvement of increasing N also gets smaller for larger N. So the best strategy would be to simply choose the largest N within the computational budget and that’s why our main experiments are conducted with $N=16$. More generally, we would expect that insights from test-time computation for LLMs / VLMs would translate similarly to this setting [1].
>
> [1] Brown, Bradley, et al. ‘Large Language Monkeys: Scaling Inference Compute with Repeated Sampling’. arXiv [Cs.LG], 2024, http://arxiv.org/abs/2407.21787. arXiv.
>
> (1/3)

---

> ### Author Response · Authors · 2024-11-20
> **Official Rebuttal to Reviewer 3SUU (2/3)**
>
> > **The computational efficiency claims, while promising, would benefit from more detailed comparisons across different model scales and task complexities. Could you provide more concrete details about the total computational resources required for training?**
>
> Due to the CUDA memory restrictions of 40G A100 that we are using, we are unable to carry out comparisons of computational efficiency with end-to-end TD-learning methods beyond 3B PaliGemma that we used in Figure 3 (left). This is because TD-learning requires keeping a separate target network as a stale copy of the critic so it uses more CUDA memory and makes distributed training harder. We are working on improving our infrastructure and applying for credits of using machines with larger CUDA memory, so would be happy to include such results in the final version of the paper.
>
> As regards to details about total computational resources, we show practical statistics in Appendix D in the updated version of the paper, which are counted on experiments done on a machine with 8 A100 GPUs. Specifically, the SFT process is standard VLM fine-tuning, which takes 20 minutes for fine-tuning a LLaVA-1.5-7b model. Getting the representations on the offline dataset takes 3 hours after vLLM acceleration. Then the critic learning takes 20 minutes and actor learning takes 30 minutes. The whole pipeline is very well optimized (at least 4x faster than original) and will be released with the final version of the paper.
>
> > **The authors don't thoroughly explore failure cases or limitations of their method.**
>
> We acknowledge that our method does have limitations, and we have conducted additional analysis to explore potential failure cases. Specifically, we calculated the success rate across different domains in the AitW Web Shopping dataset, as shown below. Our results show that success rates are notably lower on some domains compared to others. To illustrate these challenges, we provide several failure cases on the AitW Web Shopping task set in Figure 7 of the updated paper. We observe that the agent successfully navigates to the shopping homepages but fails to click the search bar after several attempts. We hypothesize that this issue arises due to a distribution shift between the pre-training data and the non-stationary environment encountered during evaluation.
>
> **Details of the new experiment.** We calculate the success rate on different domains on the AitW Web Shopping dataset, and find that the success rate on the newegg, bestbuy, and costco domains is lower than the other. **We show several failure case examples in Figure 7 of the updated version of the paper.** We observe that the agent successfully arrives at the web shopping homepage, but fails to click the search bar after several attempts. This is probably because there is a distribution shift from the pre-training data and the non-stationary environment.
>
> | **Website**  | **Success Rate** |
> |----------|--------------|
> | newegg   | 26.7         |
> | bestbuy  | 33.3         |
> | walmart  | 46.7         |
> | ebay     | 63.0         |
> | costco   | 33.3         |
>
> > **The stability analysis across different random seeds and hyperparameters could be more comprehensive.**
>
> Note that the metric of average over 3 seeds introduces a standard deviation of only 2%, which is quite low compared to many results in standard deep RL literature. That said, using 3 seeds is a compromise given the practical constraints on time, compute, and monetary budget that we have to do. We also note that this is absolutely consistent with prior works in the device control domain (Table 1 in DigiRL). We would like to note that evaluations in the device control domain are much more costly and slow compared to standard deep RL benchmarks: in fact, each evaluation involves 96 times of restarting and controlling a real Android emulator and can take more than 6 hours (more than 300 times slower interaction) on a T4 machine that we are using and costly queries into Gemini-1.5-Pro (around $1 for every 10 trajectories). We are working on obtaining more compute and Gemini credits so that we will include the results of five seeds in the final version.
>
> (2/3)

---

> ### Author Response · Authors · 2024-11-20
> **Official Rebuttal to Reviewer 3SUU (3/3)**
>
> > **From a technical perspective, while the idea of fine-tuning representations before freezing them for Q-learning is practical, it's a relatively straightforward combination of existing techniques rather than a fundamental advance in how we approach VLM training.**
>
> To the best of our knowledge, we are not aware of any prior work in device control that addresses the challenges of applying offline TD learning which has the potential of significantly improving efficiency of learning. Taking advantage of a Q-function from TD-learning is more challenging but also leads to better performances. **While it may seem relatively straightforward in hindsight, training a reliable Q-function requires careful algorithmic designs against other straightforward but not well-performing solutions. Digi-Q shows that these seemingly simple differences can make a big difference in practice.** As shown in Table 2, training a reliable Q-function requires careful algorithmic designs. For example, training Q-functions with MC return or without using capable VLMs fail to learn the relationship between the states (current screenshots) and the pixel-level actions (e.g. coordinates of tapping) with limited data (1296 trajectories). As shown in Figure 3 (left), naively fine-tuning the entire VLM backbone with TD-learning does not work either because of computational inefficiency and numerical instability. To be able to use the pre-trained capability of VLMs while avoiding the instability of fine-tuning the entire VLM backbone, we thus proposed the representation fine-tuning procedure with an appropriately chosen unsupervised objective and it turned out to be able to overcome the instabilities of TD learning to arrive at a reliable Q function.
>
> Once we have a Q function, we can optimize our policy with the Q function through sampling the actions and evaluating with Q function, opening up new possibilities of more efficient policy extraction methods such as best-of-N policy extraction that are infeasible with DigiRL. **These challenges are not studied in prior works on device control such as DigiRL where only a state-only V function is used but we have found that training a Q function that learns the relationship between states and pixel-level actions is much harder.** Given these differences, we think the improvement in terms of methodology is significant and fundamental.
>
>
> > **The ablation studies, while thorough in some areas, don't fully explore the sensitivity of the method to various design choices, particularly in the representation fine-tuning phase.**
>
> We conducted two new experiments to assess the sensitivity of the representation fine-tuning phase to the quantity of offline data. **Please let us know if there are any specific ablations you want us to add, in which case we can try to add them here.**
>
> **Experiment 1.** We performed an ablation study on the number of trajectories used in the offline dataset for the AitW Web Shopping task set. We evaluated the model's performance across three seeds for each setting when halving the number of trajectories in the offline data. The results demonstrated that the model's performance remained steady, with only a 1.5% performance difference. This suggests that the method is robust to variations in the amount of offline data, implying that Digi-Q is not that sensitive to offline data size.
>
> | Offline trajectory number | Success Rate         |
> |---------------------------|----------------------|
> | 1296 (paper setting)      | $49.7 \pm 3.5$      |
> | 512                       | $48.2 \pm 2.1$      |
>
> **Experiment 2.** We observe that the performance of Digi-Q is robust under SFT targets with different thresholds. Some examples of the thresholds between images are shown in Figure 8 in the updated version of the paper. The first transition only has a minor difference on the top left of the screen (clock time), and has a difference of $1.6$. The second transition has a major difference on the screen (search suggestions), and has a difference of $232.8$. Here we ablated on a threshold of 1, 30, and 1000. We calculate the number of yes/no targets of these thresholds, as shown in the table below. Success rate results below show that the success rates do not differ that much, demonstrating the robustness of the SFT method under different image difference thresholds.
>
> | Threshold | #Yes  | #No   | Success Rate |
> |-----------|-------|-------|--------------|
> | 1         | 13548 | 3525  | 48.1         |
> | 30        | 11633 | 5440  | 43.8         |
> | 1000      | 8284  | 8789  | 44.8         |
>
> (3/3)

---

### Official Review · Reviewer_sUX9 · 2024-11-03

**Soundness:** 3
**Presentation:** 4
**Contribution:** 3
**Rating:** 6
**Confidence:** 3

**Summary:**

This paper presents a novel method for training VLM-based RL agents. It is well known that training a VLM-based value network is highly unstable when using TD learning. This method first fine-tunes the VLM using representation learning to differentiate between actions that lead to transitions in the state space. The VLM parameters are then frozen, and the Q function on top of those layers is updated using the TD target. The policy is updated using the best of N actions from this Q function.

The main results focus on web-based navigation tasks, and the improvements are substantial. The accuracy per FLOP of training is also efficient and stable compared to fine-tuning an entire VLM.  There are extensive ablations verifying each part of the method.

**Strengths:**

* Benchmarks are sufficient; GPT-4V and Gemini are both s strong benchmarks.
* Qualitative visualizations strongly support the hypothesis.
* The ablations are numerous, exploring different representation learning methods to train the Q function, comparisons with Monte Carlo learning, and divergence from the behavior policy.
* The paper is extremely clear and well written.
* Experimental evaluations are strong and address difficult web navigation tasks.

I recommend acceptance of this work, it presents strong results in an area of RL that is of very high impact currently.  Using VLMs to perform RL tasks is currently a direction of interest to most of the RL community.

**Weaknesses:**

* Novelty is a bit lacking, the main contribution of this method is simply fine-tuning upon the frozen lays using the TD loss after representation learning.  This is especially apparent put into the context of DigiRL

**Questions:**

This paper is very well written and does not warrant any immediate questions.

---

> ### Author Response · Authors · 2024-11-20
> **Official Rebuttal to Reviewer sUX9 (1/1)**
>
> Thank you for your thoughtful review and a positive assessment of this paper! To address your concerns, we emphasize that our work tackles the problem of applying offline value-based RL to device control tasks, a problem that has been largely overlooked in prior work, including DigiRL. We show results that outperform DigiRL. To strengthen our performance improvements, we provide additional results, comparisons, and detailed clarifications regarding the methodological advancements of our approach.
>
> **Please let us know if these responses address your concerns and if so, we would be grateful if you would be willing to raise your score. We remain available for further discussion.** Below, we address your points in detail:
>
> > **Novelty is a bit lacking, the main contribution of this method is simply fine-tuning upon the frozen layers using the TD loss after representation learning. This is especially apparent put into the context of DigiRL**
>
> To the best of our knowledge, we are not aware of any prior work in device control that addresses the challenges of applying offline TD learning which has the potential of significantly improving sample efficiency.
>
> On the other hand, **DigiRL only trains a state-only value function $V(s)$ by regressing against Monte-Carlo return estimates.** Taking advantage of a Q-function from TD-learning, as what we do in Digi-Q, is more challenging but also leads to better performances. As shown in Table 2, training a reliable Q-function requires careful algorithmic design. For example, training the Q-function with MC return or without using capable VLMs fails to learn the effect of pixel-level actions (e.g. coordinates of tapping) on states (current screenshots), especially with limited data (1296 trajectories). As shown in Figure 3 (left), naively fine-tuning the entire VLM backbone with TD-learning does not work either because of computational inefficiency and numerical instability. To be able to use the pre-trained knowledge in the VLM while avoiding the instability of fine-tuning the entire VLM backbone, we thus proposed the representation fine-tuning procedure with an appropriately chosen unsupervised objective and it turned out to be able to overcome the instabilities of TD learning to arrive at a reliable Q function.
>
> **Training a Q-function, $Q(s, a)$ opens up new possibilities for training policies.** For example, we show that we can optimize our policy by sampling multiple actions, evaluating the Q-value on all of them, and picking the best one. This makes more use of test-time compute and performs better (as we show in Table 1 of the paper). Note that this kind of a more effective policy extraction method is infeasible with a value function that DigiRL uses. Given these differences, we think the improvement in terms of methodology is significant and fundamental.
>
> While these methodological changes look simple, they lead to an 9.9% relative improvement over DigiRL (which is substantial), our closest and strongest baseline on this problem.
>
> (1/1)

---

### Official Review · Reviewer_BK8U · 2024-11-04

**Soundness:** 3
**Presentation:** 3
**Contribution:** 3
**Rating:** 6
**Confidence:** 4

**Summary:**

This paper develops a new RL method called Digi-Q. It addresses how to effectively use value-based offline reinforcement learning (RL) to train visual-language model (VLM) agents in dynamic environments (such as mobile device control) by training on frozen intermediate layer representations of the VLM through temporal difference (TD) learning instead of training on the entire VLM.
I think the main innovation is Digi-Q. According to the paper, they address a series of challenges brought about by large-scale value-based offline policy RL: training instabilities associated with running temporal difference (TD) learning on large models, and inefficiency of TD backup per unit of computation. If you solved any of them, I will think it's a good novelty.

**Strengths:**

1. The article is well-structured, with references for every sentence. And the author didn't speculate on the causes of certain phenomena observed in the experiments, without having conducted the experiments or found relevant references in the literature.
2. It solves a series of challenges brought about by large-scale value-based offline policy RL: training instabilities associated with running temporal difference (TD) learning on large models, and inefficiency of TD backup per unit of computation. It ensures innovation.
3. It also gives annotations of the papers from which ideas were borrowed, so that reviewers will not always think of some familiar operations when reading the experimental part, and spend a lot of time to confirm whether there is plagiarism.
4. All research is based on the latest literature

**Weaknesses:**

1. Overfitting and Catastrophic Forgetting
2. Misclassification of Visual Changes
3. Real-World Applicability
4. General Limitations and Feedback
Check the Questions to see the details, thanks :)

**Questions:**

In the paper, you mentioned that Digi-Q first fine-tunes representations of a VLM with a binary classification objective to enable it to pay attention to actionable features of an input scene. The sample step is in Figure 9 and it looks like targeted. How do you solve the Overfitting and Catastrophic Forgetting problem? Can you try to do some case studies or use different datasets to prove your performance improvement?
You mentioned that observation is that in device control problems, a useful action should lead to a substantial visual change in the pixel values of a scene (e.g., successfully typing a search query and pressing enter on google.com should cause the page to change substantially to now show a list of search results, whereas an unsuccessful search attempt will change none to few pixels of the original scene). But in my daily life, the more possible is that a failed attempt can also lead to a substantial visual change. For a very common example, the page changes to whole white or 404 at my iPhone App when you fresh the web page but lose internet. Can you provide some statistical evidence for your observation? I think your assumption is to decline your model in generalization ability again. I know what Reinforcement Learning is but sometimes it's not normal in the real world.
Can you talk more about limitation? Do you ask some people to test your model and do some case studies? Do get some feedback? Like you said before, you have some observations, so I assume it's only only from you but also from your tester's feedback.

---

> ### Author Response · Authors · 2024-11-20
> **Official Rebuttal to Reviewer BK8U (1/3)**
>
> Thank you for your valuable review and feedback on our paper. We have conducted additional experiments and added clarifications to address the raised concerns. These include clarifying our approach to mitigate overfitting and catastrophic forgetting, providing statistical evidence for the motivation for representation fine-tuning and analysis results on the limitations. We have updated the paper accordingly (changes highlighted in blue) and included further ablation studies, case analyses, and explanations to strengthen the evaluation of our method. Below, we address each of your concerns in detail:
>
> > **Digi-Q first fine-tunes representations of a VLM with a binary classification objective to enable it to pay attention to actionable features of an input scene. The sample step is in Figure 9 and it looks targeted.**
>
> > **How do you solve the Overfitting and Catastrophic Forgetting problem? I think your assumption is to decline your model in generalization ability again.**
>
> We think there might be some misunderstanding on our part regarding the reference to "overfitting" and "catastrophic forgetting", so please let us know if the response below does not answer your concerns. Our interpretation of this term was in regards to forgetting the VLM abilities due to RL fine-tuning on limited device control data. We find that this is not the case, and the VLM policy produced by Digi-Q is still able to effectively solve new challenging tasks from new initial states. Reducing the amount of data to half also does not substantially reduce the success rate of the Digi-Q policy as shown below, indicating overfitting is not a concern.
>
> **Details of the new experiment**. Specifically, we ablated the number of trajectories in the offline dataset for the Web Shopping task set, using three seeds for each setting. The results demonstrate steady performance, with only a 1.5% difference when the number of trajectories is halved. This indicates the method’s robustness to variations in data quantity and underscores its effectiveness in the targeted domain.
>
> | Offline trajectory number | Success Rate         |
> |---------------------------|----------------------|
> | 1296 (paper setting)      | $49.7 \pm 3.5$      |
> | 512                       | $48.2 \pm 2.1$      |
>
>
> > **Sub-question 2: Can you try to do some case studies to prove your performance improvement?**
>
>
> Yes, we did some case studies to compare the performance difference of the critic trained with and without representation fine-tuning as shown in Figure 4 in the paper. We found that qualitatively the critic with our representation fine-tuning procedure indeed assigns more accurate advantage values than the critic without. We also did a case study in Figure 5 showing that DigiQ can effectively learn optimal behaviors through “stitching” suboptimal trajectories. **If there is a particular experiment or ablation that you think will help prove the performance improvement even more, we are happy to add it if you have suggestions.**
>
> (1/3)

---

> ### Author Response · Authors · 2024-11-20
> **Official Rebuttal to Reviewer BK8U (2/3)**
>
> > **But in my daily life, the more possible is that a failed attempt can also lead to a substantial visual change. For a very common example, the page changes to whole white or 404 at my iPhone App when you fresh the web page but lose internet. Can you provide some statistical evidence for your observation?**
>
> The offline data is collected by AutoUI, which is a pre-trained policy model on the Android device control domain. We have collected statistics showing that most transitions are good if there is a large euclidean distance between the two images, as shown below in the new experiments for statistical evidence of our motivation. To provide statistical evidence for the observation that a successful attempt can often lead to substantial visual changes.
>
> In regards to uncontrolled transitions akin to what you mentioned, we remark that our offline dataset does include many examples that are just irrelevant towards solving the task, like RECAPTCHA. These does not affect the quality of representation and TD-learning (as we see from the success of Digi-Q). We hypothesize that this is because representation fine-tuning here simply tries to make the VLM aware of “what’s changing in scene”, not “whether the change is good or not”. “whether the change is good or not” is what a Q-function attempts to model anyways, and it can do so well as long as the representation of VLM does provide some features for useful changes to the scene. We have updated this in the paper (page 6, footnote 1).
>
> **Details on the new Experiment.** We sample a subset of 50 offline trajectories (around 500 transitions in total). We label the transitions with euclidean distance larger than a threshold as positive, else negative. We also manually label whether a transition is effective towards its task goal. If it’s effective we label it positive, else negative. Then we calculate the agreement accuracy and get 74.5%, where a random prediction will only yield 50%. Instead of directly using it to make a prediction of the attempt being successful or not, this is just a simple objective that we use to train the VLM to pay attention to the relation between and action and the screen to induce action-aware representations.
>
> (2/3)

---

> > ### Comment · Reviewer_BK8U · 2024-11-27
> > **Response to 2/3**
> >
> > The agreement accuracy of 74.5% is a good number but not enough. I think you should show this impact on your final performance score. But thanks for your new experiments.

---

> ### Author Response · Authors · 2024-11-20
> **Official Rebuttal to Reviewer BK8U (3/3)**
>
> > **I know what Reinforcement Learning is but sometimes it's not normal in the real world. Can you talk more about limitation? Do you ask some people to test your model and do some case studies? Do get some feedback?**
>
> Thanks for the suggestion! We have updated the paper to better clarify limitations and include a discussion about case studies in Section 6. As you pointed out, reinforcement learning can sometimes be impractical in real-world settings. For example, in robotic manipulation, defining a reward function often requires specialized tools like mocap systems, while trial-and-error interaction can be unsafe, costly, and time-intensive. These challenges limit the direct applicability of reinforcement learning. In response to these limitations, we rely on offline RL, which eliminates the need for real-world interaction by learning policies from static, pre-collected data. Offline RL is particularly suitable for our device-control problem, where incorrect actions could lead to time-consuming or unsafe outcomes. However, scaling offline RL to large VLMs introduces additional challenges, which we address with Digi-Q, our proposed agent that trains a VLM-based Q-function for device control tasks.
>
> While we have not conducted direct case studies involving user feedback, our experiments serve as a form of evaluation, shedding light on the performance of our method across different domains. For instance, in the AitW Web Shopping dataset, we observe lower success rates in certain domains such as Newegg, BestBuy, and Costco compared to others. Figure 7 in the updated paper illustrates failure cases in the AitW Web Shopping subset: the agent successfully navigates to the shopping homepage but fails to click the search bar after several attempts. This likely stems from a distribution shift between pre-training data and the non-stationary environment. Moving forward, incorporating real-world case studies and user feedback could further validate and refine our approach. These insights would complement our experimental findings and help address practical challenges more comprehensively.
>
> **Details of the new Experiment.** We calculate the success rate on different domains on the AitW Web Shopping dataset, and find that the success rate on the newegg, bestbuy, and costco domains lower than the other. We show several failure case examples on the AitW Web Shopping task set in Figure 7 of the updated version of the paper. We observe that the agent successfully arrives at the web shopping homepages, but fails to click the search bar after several attempts and very few trajectories on those websites can successfully perform searching. This shows that although DigiQ can significantly strengthen the performance of the pre-trained agent on websites the agent is familiar with, its improvement can be less significant if the task is too out-of-distribution for the pre-trained agent.
>
> | **Website**  | **Success Rate** |
> |----------|--------------|
> | newegg   | 26.7         |
> | bestbuy  | 33.3         |
> | walmart  | 46.7         |
> | ebay     | 63.0         |
> | costco   | 33.3         |
>
> (3/3)

---

> > ### Comment · Reviewer_BK8U · 2024-11-27
> > **Thanks your response to 3/3**
> >
> > I like your response. As I said before, I like your writing style. And I think score 6 is fair enough.

---

> ### Comment · Reviewer_BK8U · 2024-11-25
> **Thanks for your response.**
>
> Hi,
> Thanks for your response. I will give you some reasons this afternoon in PST time. Sorry for the delay. A little busy right now.

---

> ### Comment · Reviewer_BK8U · 2024-11-27
> **Responses to 1/3**
>
> Hi, maybe my words were not clear the first time. I mean your title is Device-Control Agents.
> More specific questions here.
> Have you conducted experiments on different types of scenarios?
> Have you verified that reducing the amount of data also changes the diversity of data distribution?
> Have you tested the model's performance on low-frequency tasks or extreme scenarios?
>
> Lack of research on the impact of other dimensions (such as data noise, and quality differences) on model performance

---

> > ### Author Response · Authors · 2024-12-01
> > **Follow-up discussion towards Reviewer BK8U**
> >
> > Thanks a lot for getting back on this! To address your concerns, we would like to kindly note that we’re running on the broadest possible set of scenarios on the AitW task set that is consistent with the prior work, DigiRL. There are also some other subsets except the General and Webshop subsets in AitW (like app installation), but app installation tasks causes security reasons because an account is needed, and tasks in the the Single task set fails to examine multi-step challenges that we’re interested in.
> >
> > To elaborate, our experiment setup and evaluation tasks are identical to DigiRL, because the focus of this work is to develop a better RL algorithm for device control instead of a generalist model checkpoint. Thus, the reason that we don’t include these tasks is the same as DigiRL (see Appendix A.1 Paragraph 1 in the DigiRL ArXiv paper):
> >
> > “ *The Android in the Wild (AiTW) task set is a large-scale dataset for android device control, containing five subsets: GoogleApps, Install, Web Shopping, General, and Single, where we select the General and Web Shopping subsets. Single subset is not considered here because all tasks in Single can be completed within one step and thus this subset fails to examine the multi-step challenges that we are interested in this paper. Install and GoogleApps are not considered due to security reasons as those tasks require an active Google account and parallel emulations can flag security concerns.*”

---

### Official Review · Reviewer_h5me · 2024-11-04

**Soundness:** 2
**Presentation:** 1
**Contribution:** 2
**Rating:** 3
**Confidence:** 4

**Summary:**

The paper proposes an approach, called Digi-Q, for learning useful behaviour for device-control by leveraging offline data and VLMs. The authors highlight the current difficulties of learning using temporal difference learning and large retrained models. To address this difficulty, the authors propose to pretrain the VLM with in-domain data with state and action pairs, together with labels indicating whether the resulting state has changed significantly after the taken action. Additionally, the authors propose to use a best-of-N action sampling strategy, where the best-of-N is calculated through an approximate Q function. On the Android-in-the-Wild (AitW) domain, experiments show that Digi-Q improves upon previous approaches. The authors also present compute efficiency comparisons and ablation studies on some of the choices within the algorithm.

**Strengths:**

The paper builds on recent algorithms and evaluates on domains that are increasingly important in the space of decision making and AI agents. The paper also highlights some of the challenges associated with reinforcement learning and VLM (e.g. instabilities that appear in practice) and proposes an approach that seems to present favorable results in experiments. Some of the ablations also give reasonable answer to important questions, for example the number of actions in the best-of-N sampling strategy.

**Weaknesses:**

The paper is generally not very rigorous from a scientific point of view. There are numerous descriptions of problems, hypothesis as to why one things are not working, and empirical justifiications that are unsubstantiated. For example the negative gradient hypothesis being mentioned multiple times originates from a paper on a preference fine-tuning, not agentic tasks. Other examples include the whole motivation in section 4.2: "REINFORCE [...] is brittle with off policy data"  "negative gradient [...] means careful tuning of learning rates must be done" ( "AWR is quite conservative and slow". When reading such statements, together with a complete misunderstanding of the fact that value-based methods do not equal off-policy learning, which also does not equates to offline learning (see Introduction), indicates to me that there is limited understanding and insights into what's happening, and therefore the contribution of the paper is lessened.

Arguably, the text could be fixed and made more precise, however these issues also arise in the algorithms and experiments. The proposed method, mixes quite a few things together: using the ArCHer learning rules, pretraining on in-domain data and best-of-N action sampling. For each of these choices, there is far from enough evidence to understand what is its importance.

Consider pretraining on in-domain data, the paper mentions that labels are created when s_{t+1} is significantly different from s_t using the l_2 distance. Is this a general or even reasonable objective? Does this assume that the environment is entirely controllable by the agent, or deterministic? Given many papers in the RL literature, it clearly does not seem to be the case. Also, there is no study on the sensitivity to value of the threshold epsilon for calculating labels.

Also, looking at Figure 3 right, we see that when the number of actions for best-of-N is set to 1, the performance is similar to Filtered BC. Why do we not see performance difference given the fact the Digi-Q is built on pretraining the VLM first?

The main results raise a few questions. First, only 3 seeds are being used, please see the numerous papers that indicate that this a bad practice [1, 2]. Second, why is the performance of DigiRL different than the one reported in the original paper? Third, concerning the ablation in Table 3, how is the performance of AWR so low? Is the procedure for AWR not the same as the one proposed in DigiRL?

The results on compute efficiency conflate a few things together. Finetuning whole LLMs with RL can be troublesome (although it is possible as reported in a few recent papers), but this problem is mixed with compute efficiency. If performance degrades in the reported experiments, and full fine-tuning brings does not give as high a score as partial fine-tuning, it has little to do with compute efficiency, but rather with the practical challenges of full fine-tuning. In this sense, it is a bit meaningless to claim that partial fine-tuning is more efficient than full fine-tuning, if the used update rules don't work with full fine-tuning.

**Questions:**

Why use a separate policy network from the value? This is mentioned on the way, but never explained or referred.

In DigiRL, the authors perform a curriculum over tasks, is this strategy also employed here?

Throughout the paper, the best-of-N strategy is referred to as being novel. I do not care if a method is novel or not, but proposing a method that is not novel (it is an incremental improvement on AWR, filtered BC and BCQ [3]]), and referring to it as being novel is not great.

[1] Deep Reinforcement Learning at the Edge of the Statistical Precipice, Agrawal et al., 2022
[2] Deep reinforcement learning that matter, Henderson et al., 2018
[3] Off-Policy Deep Reinforcement Learning without Exploration, Fujimoto et al., 2018

---

> ### Author Response · Authors · 2024-11-20
> **Official Rebuttal to Reviewer h5me (1/4)**
>
> Thank you for your review and feedback on our paper. To address the concerns regarding terminology and motivation, we have updated the paper (changes shown in blue) to make wording and contributions more precise and motivations more clear. We provide **additional results** ablating each of the design decisions in Digi-Q. We also clarify below that the inconsistencies between the Digi-RL paper and our results for the Digi-RL approach stems from a difference in the offline dataset used for training, and non-stationarity of the web environment. We also clarify some details about baselines and certain ablations that we believe are already present in the paper.
>
> **Please let us know if your concerns are addressed, and if so, we would be grateful if you are willing to increase your score. We are happy to discuss further**  We answer your questions below:
>
> **The negative gradient hypothesis being mentioned multiple times originates from a paper on a preference fine-tuning, not agentic tasks. The whole motivation in section 4.2 of ,"REINFORCE [...] is brittle with off policy data", and "negative gradient [...] means careful tuning of learning rates must be done" are not supported.**
>
> We note that this reference actually discusses negative gradients in the context of reward optimization (see Equation 3.5 in Tajwar et al. ICML 2024) once a reward function is extracted from preferences although indeed their experiments were largely performed on simulated preference-optimization tasks with a known reward function. Hence we believe that these claims should in principle not be limited to preference optimization, but apply to reward optimization in general.
>
> That said, we have now updated the paper to forward reference our own experiments with REINFORCE which show the instability issue with negative gradient. A comprehensive analysis of the negative gradient effect in the reasoning domain was also carried out in Section 5.7 from ArCHer [2] (see line 298 in pdf) which focuses on agentic tasks and they found similar conclusions.  We also observed a similar conclusion in our experiment results of REINFORCE in Table 3. We have edited the paper in Section 4.2 to refer to Table 3 for this hypothesis.
>
>
> We have also updated the text in Section 4 to avoid the impression of overclaim by removing any statements that are not absolutely clear from the aforementioned evidence.
>
> > **"AWR is quite conservative and slow" is not supported.**
>
> Thanks for the question! To clarify, by “conservative and slow” we mean that AWR does not train the VLM policy to deviate far away from the dataset policy. In Table 3 of the submission, we already measure the KL divergence between the policies learned by Digi-Q and AWR and the behavior policy. AWR attains a very low divergence justifying this. We note that a similar conclusion has been made in Figure 9 (Left) of [2] and Figure 1 of [5].
>
> That said, to avoid any confusion or misunderstanding due to imprecise terminology, we have now edited the paper to precisely identify what we mean by conservative (i.e., “conservative in the sense of small divergence from the behavior policy”) and removed the word “slow”.
>
> (1/4)

---

> ### Author Response · Authors · 2024-11-20
> **Official Rebuttal to Reviewer h5me (2/4)**
>
> > **The proposed method mixes quite a few things together and does not disentangle the effects of different factors. using the ArCHer learning rules, pretraining on in-domain data and best-of-N action sampling. For each of these choices, there is far from enough evidence to understand what is its importance.**
>
> We agree that ablation studies are very important, and as a result we already presented ablations in Table 2, 3 and Figure 3 (right) of the submission, as we justify below. We have added new experimental results to solidify ablation experiments as we discuss below. Please let us know if you think specific ablation studies would be useful, and we are happy to add them.
>
> - **ArCHer learning rules.** By comparing DigiQ w/ MC return and DigiQ in Table 2, we show the effectiveness of the ArCHer learning rule for learning the value function against learning the value function with MC return.
> - **Representation fine-tuning.** By comparing DigiQ w/ Off-the-shelf VLM and CLIP+BERT with Digi-Q, we show the effectiveness of using the representation fine-tuning procedure prior to RL. We also add an ablation over different ways of fine-tuning the representation (more on this in response to your next question) and find that our approach still performs best.
> - **Best-of-N policy extraction.**: By comparing DigiQ with different policy extraction methods in Table 3 and the effect of the number of actions in Figure 3 (right), we have shown that Best-of-N achieves the best balance between policy improvement and conservatism as measured by KL divergence.
>
> Overall, the use of representation fine-tuning and ArCHer update rule contribute to training a reliable Q function while best-of-n policy extraction makes best use of this Q function compared to the alternatives. All components work together to ensure the effectiveness of DigiQ. **Please let us know if certain specific ablation studies are required.**
>
> > **looking at Figure 3 right, we see that when the number of actions for best-of-N is set to 1, the performance is similar to Filtered BC. Why do we not see performance difference given the fact the Digi-Q is built on pretraining the VLM first?**
>
> We would like to mention that both Digi-Q, Filtered BC, and DigiRL use the same policy network from the pre-trained checkpoint of AutoUI to keep a fair comparison. The representation fine-tuning procedure is only conducted for using the VLM representations for the critic. As explained in response to AWR and the ablations above, while the use of a pre-trained VLM and representation fine-tuning procedure can train a good Q function, best-of-N training with N set to 1 does not make any sufficient use of the Q-function since it reduces down to simply imitating a high-advantage action from the behavior policy (i.e., filtered BC) , hence resulting in an inferior performance compared to using more actions in best-of-N. On the other hand, Filtered BC simply imitates all actions in successful trajectories without depending on a learned Q function, thus following a very different update rule with DigiQ with best-of-N set to 1 that uses a Q function.
>
> > **Consider pretraining on in-domain data, the paper mentions that labels are created when s_{t+1} is significantly different from s_t using the l_2 distance. Does this assume that the environment is entirely controllable by the agent, or deterministic? Is this a general or even reasonable objective?**
>
> We clarify that our proposed representation learning objective is specific to pixel-level device control problems, where an ineffective action usually clicks on non-interactive elements, e.g. clicking on some random text or a blank space. These actions will not lead to any progress towards solving the task. We utilize this feature of device control problems and make VLMs learn to distinguish whether there will be a transition or not.
>
> We do not intend to claim that this objective is general or will work for any control problem, and we have updated the text in Section 4 to explicitly reflect this. We also note that our goal is not to develop the best possible representation learning objective either, but to find one that is simple (given current VLMs that can only take in one input image) but is able to prime the VLM for TD-learning. We succeed towards this goal since Digi-Q attains SoTA performance.
> Of course, there might be other objectives that perform better and are more generally applicable, but developing such objectives is orthogonal to our contribution.
>
> (2/4)

---

> ### Author Response · Authors · 2024-11-20
> **Official Rebuttal to Reviewer h5me (3/4)**
>
> > **there is no study on the sensitivity to value of the threshold epsilon for calculating labels.**
>
> **Experiment 1.** We observe that the performance of Digi-Q is robust under SFT targets with different thresholds. Some examples of the thresholds between images are shown in Figure 8 in the updated version of the paper. The first transition only has a minor difference on the top left of the screen (clock time), and has a difference of $1.6$. The second transition has a major difference on the screen (search suggestions), and has a difference of $232.8$. Here we ablated on a threshold of 1, 30, and 1000. We calculate the number of yes/no targets of these thresholds, as shown in the table below. Success rate results below show that the success rates do not differ that much, demonstrating the robustness of the SFT method under different image difference thresholds.
>
>
> | Threshold | #Yes  | #No   | Success Rate |
> |-----------|-------|-------|--------------|
> | 1         | 13548 | 3525  | 48.1         |
> | 30        | 11633 | 5440  | 43.8         |
> | 1000      | 8284  | 8789  | 44.8         |
>
> **Experiment 2.** We also add a new experiment where we sample a subset of 50 offline trajectories (around 500 transitions in total). We label the transitions with euclidean distance larger than a threshold as positive, else negative. We also manually label whether a transition is effective towards its task goal. If it’s effective we label it 1, else zero. Then we calculate the agreement accuracy and get 74.5%. Note that this is just a simple objective that we use to train the VLM to induce action-aware representations. In the end we still evaluate using the success rate.
>
> > **The main results raise a few questions. only 3 seeds are being used, please see the numerous papers that indicate that this a bad practice**
>
>
> Using 3 seeds is a compromise given the practical constraints of compute, wall-clock time and monetary budget (querying the Gemini 1.5 Pro API). It is also consistent with prior works in the device control domain (Table 1 in DigiRL). We would like to note that evaluations in the device control domain are much more costly and slow compared to experiments on standard deep RL benchmarks such as MuJoCo and Atari, where each evaluation involves **96 times** restarting and controlling a real Android emulator can take more than 6 hours (more than 300 times slower than interactions on MuJoCo and Atari) on a machine equipped with a T4 GPU. Each evaluation rollout also queries a Gemini-1.5-Pro model (around $1 for every 10 rollouts). Additionally, the size of our 7B critic  is more than 1000 times larger than typical 3-layer convolutional neural networks used in MuJoCo and single-task Atari (with fewer than 7M parameters). We are working on obtaining more compute and Gemini API credits so that we can try to run more seeds (e.g., 5 seeds) and plan to include the results of more seeds in the final version.
>
> > **why is the performance of DigiRL different than the one reported in the original paper**
>
> While we used the public DigiRL repo for reproducing their results, the non-stationary nature of device control problems result in slight differences in numbers  from DigiRL. As mentioned in Section 3 of DigiRL, the environment for device control is non-stationary by nature because of the interactions with the ever-changing real Internet (i.e. websites have changed from the time when DigiRL was evaluated), where the performance of the same model checkpoint can change. For e.g., our reproduction results in abetter performance for DigiRL (averaged 49.8% across task slices) compared to original results in DigiRL (averaged 48.7% across task slices).  Of course, the gap for DigiRL is only 1-2% over changes in websites, but this gap could be larger for simple baselines without fine-tuning like AutoUI (upto 5%).
>
> The reason why only DigiRL numbers appeared different was because we chose to re-evaluate DigiRL only (as opposed to re-running all baselines). This was because we did not have unlimited compute and monetary budget for evaluations and had to compromise towards only re-evaluating the closest and the strongest baseline (DigiRL), while retaining numbers for the others directly form prior work. For the final version, we will re-run all baselines.
>
> (3/4)

---

> ### Author Response · Authors · 2024-11-20
> **Official Rebuttal to Reviewer h5me (4/4)**
>
> > **Value-based methods do not equate off-policy methods and do not equate offline methods.**
>
> Thanks for the pointer. We are of course aware of this difference and many sentences in our submission pdf do already reflect this (e.g. the title of the submission itself refers to value-based offline RL only; in line 117-118, we mentioned “In traditional RL, off-policy and offline RL algorithms that train a state-action value function (i.e., a Q-function) via temporal-difference learning (TD-learning) are known to be substantially more sample efficient and effective” with the awareness that off-policy and offline RL algorithms are different and a subset of algorithms in off-policy and offline RL algorithms). That said, wordings in some places of the submission might have been imprecise and we have now updated the paper to address it. For example, we have updated the wording in Line 76 that “Digi-Q is handling the challenges of value-based offline RL only”. We are happy to address any specific wording issues that you notice. Please let us know if you would like to make changes elsewhere too.
>
> > **concerning the ablation in Table 3, how is the performance of AWR so low? Is the procedure for AWR not the same as the one proposed in DigiRL?**
>
>
> We note that Digi-RL prescribes an improved policy extraction procedure that relies on doubly robust estimator and MC return compared to AWR policy extraction. Figure 9 of DigiRL also shows that vanilla AWR may not be a strong baseline in device control problems as it learns much more slowly compared to other methods.
>
> > **Why use a separate policy network from the value? This is mentioned on the way, but never explained or referred.**
>
>
> This is to keep a fair comparison with DigiRL which uses the AutoUI[4] checkpoint for the policy. We would like to control the variable to show that the improvement of DigiQ comes from a better critic and policy extraction methods, instead of a more capable pre-trained actor.
>
> > **In DigiRL, the authors perform a curriculum over tasks, is this strategy also employed here?**
>
> The curriculum over tasks is only used in the online phase of DigiRL instead of the offline phase to improve the efficiency of online learning. We focus on the offline setting so it is more preferable to make best use of all the offline data to maximize sample efficiency.
>
> > **I do not care if a method is novel or not, but proposing a method that is not novel is not good.**
>
>
> We have tuned down our claim on the novelty of best-of-n training. For example, we avoided describing the best-of-n training as a “novel” approach on line 228 of the updated version of the paper.
>
> [1] Bai, Hao, et al. "DigiRL: Training In-The-Wild Device-Control Agents with Autonomous Reinforcement Learning." Advances in Neural Information Processing Systems 37 (NeurIPS 2024).
>
> [2] Zhou, Yifei, et al. "ArCHer: Training Language Model Agents via Hierarchical Multi-Turn RL." Proceedings of the 41st International Conference on Machine Learning (ICML 2024).
>
> (4/4)

---

> > ### Comment · Reviewer_h5me · 2024-11-25
> >
> > Dear Authors, I appreciate your rebuttal, but I am still left concerned about the submission. There are some red flags that have been left unaddressed, like the number of seeds and the fact that some baselines have not been properly re-ran (and I can't even find a mention of this in the original submission). Given that the environment is constantly changing, it is just another sign of how important it is to have robust evaluations, and I would argue that even 5 seeds is not enough.
> >
> > The answers provided around negative gradient term, the AWR update rule, and the representation learning method are, to me, hand-wavy and not properly evaluated by the experiments. Some concerns have been completely unanswered, like the claims on compute efficiency. For some other concerns the authors mention that they have addressed them, but, as an example, the claims of novelty can still be found throughout the paper.
> >
> > I would encourage the authors to situate their claims a bit better within the RL literature (for example, there are alternatives to using MC that do not equate to AWR or some of the update rules presented by Archer) , which would drive experiments that more strongly support the main contributions, that is, a quite specific pre-training representation learning phase and the use of best-of-N. As a result, I unfortunately can't see myself upgrading my score.

---

> > > ### Author Response · Authors · 2024-12-01
> > > **Follow-up discussion towards Reviewer h5me**
> > >
> > > Thanks a lot for getting back to us! To address the remaining concerns, we promise to run more seeds for each method for the submission. We have been trying to run more seeds for each method since the beginning of the rebuttal period, and even now, but are running into compute cost bottlenecks ($1000 for running Gemini-based evaluations, cloud platforms for compute), and due to proximity to end of the year, it has been challenging to get compute donations for these baselines. In addition, please do note that the closest prior work that has been accepted at NeurIPS conference, DigiRL, also did their evaluation on 3 seeds. Since Digi-Q attains 9.9% relative improvement over DigiRL, we believe that analogous to this prior work, 3 seeds should provide for a meaningful signal for showing efficacy of Digi-Q.
> > >
> > > **Re-running baselines:** That said, with the limited compute quota we could get, we have rerun several baselines, especially the ones based on prompting strong proprietary models and UI agents, as shown in the Table below. In summary, we find that all baselines perform similarly to the numbers in the DigiRL paper. For example, on the AitW General set, there is a ~2% difference in success rate for the Set-of-Marks (Gemini-1.5-Pro) method, and ~5% difference in success rate for the CogAgent method performing worse than what is reported in the DigiRL paper. That said, please do note that baseline performance for prompting-based methods is expected to vary from time to time: most of these baselines involve proprietary model checkpoints that has a temperature, and keep on evolving. We want to reiterate that our intention here in the paper was not to hide numbers or selectively rerun baselines, we simply made a logical compromise to run the most promising baseline (which involved training) as opposed to prompting-based methods or methods based on off-the-shelf models in the submission. Since our latest numbers for these methods are largely worse, this implies that DigiQ is still the most performant method and no conclusions will change.
> > >
> > >
> > > | Method                        | AitW General (Train) | AitW General (Test) |
> > > | ----------------------------- | -------------------- | ------------------- |
> > > | Set-of-Marks (Gemini-1.5-pro) | 32.3/**30.2**        | 16.7/**14.6**       |
> > > | CogAgent                      | 25.0/**18.8**        | 25.0/**29.5**       |
> > >
> > > *Bolded numbers are results we we-ran during the rebuttal period. Unbolded numbers are original results from the DigiRL paper.*
> > >
> > >
> > > > the claims of novelty can still be found throughout the paper
> > >
> > > We apologize for any oversight here. We did find an additional claim of novelty in the Related Works and abstract, and we will remove them in a later version of the paper. Is there a particular claim you would want us to remove? We are flexible about this and pretty open to addressing these.

---

### Author Response · Authors · 2024-11-23
**Happy to engage in further discussions**

Dear reviewers,

Thanks so much for your feedback on the paper. As the discussion is coming to an end, please let us know if our additional experiments and clarifications have addressed your concerns. We are happy to engage in further discussions.

---

### Author Response · Authors · 2024-11-24
**Follow-up on Rebuttal Feedback**

Dear Reviewers,

Thank you for taking the time to review our work. We greatly appreciate the effort you’ve put into providing thoughtful feedback.

As the discussion phase draws to a close, we wanted to follow up regarding our responses to your comments. We have worked diligently to address all the concerns raised and hope that our revisions demonstrate the merit of our paper.

We understand you may have a busy schedule, but we would greatly appreciate any additional feedback on our rebuttal, or confirmation if our responses have resolved your concerns. If there are any lingering issues, we would be happy to address them promptly.

Thank you again for your valuable time and insights throughout this process.

Best regards,

Authors

---

### Meta-Review · Area_Chair_GF4T · 2024-12-19

**Metareview:**

The reviewers generally view this paper as presenting an incremental, implementation-focused contribution, with respect to the prior work, Digi-RL. However, the clear empirical design and discussion, combined with the demonstrated improvements in the domain of computer use make this a useful contribution to the field. The Digi-Q recipe, which the authors thoroughly studied in this work, can benefit future researchers in this area.

**Additional Comments On Reviewer Discussion:**

Reviewer h5me asked for improvements to the exposition and both h5me and NJc5 requested additional experimental details, which I believe the authors sufficiently addressed in their rebuttal.

---

### Decision · Program_Chairs · 2025-01-22

Accept (Poster)